# A Simple Approach to Automated Spectral Clustering

**Jicong Fan**[1,2], **Yiheng Tu**[3,4], **Zhao Zhang**[5]*, **Mingbo Zhao**[6], **Haijun Zhang**[7]

[1]The Chinese University of Hong Kong, Shenzhen   [2]Shenzhen Research Institute of Big Data
[3]Chinese Academy of Science, Beijing   [4]University of Chinese Academy of Sciences, Beijing
[5]Hefei University of Technology, Hefei   [6]Donghua University, Shanghai
[7]Harbin Institute of Technology, Shenzhen
fanjicong@cuhk.edu.cn   yihengtu@gmail.com   cszzhang@gmail.com
mzhao4@dhu.edu.cn   hjzhang@hit.edu.cn

## Abstract

The performance of spectral clustering heavily relies on the quality of affinity matrix. A variety of affinity-matrix-construction (AMC) methods have been proposed but they have hyperparameters to determine beforehand, which requires strong experience and leads to difficulty in real applications, especially when the inter-cluster similarity is high and/or the dataset is large. In addition, we often need to choose different AMC methods for different datasets, which still depends on experience. To solve these two challenging problems, in this paper, we present a simple yet effective method for automated spectral clustering. First, we propose to find the most reliable affinity matrix via grid search or Bayesian optimization among a set of candidates given by different AMC methods with different hyperparameters, where the reliability is quantified by the *relative-eigen-gap* of graph Laplacian introduced in this paper. Second, we propose a fast and accurate AMC method based on least squares representation and thresholding and prove its effectiveness theoretically. Finally, we provide a large-scale extension for the automated spectral clustering method, of which the time complexity is linear with the number of data points. Extensive experiments of natural image clustering show that our method is more versatile, accurate, and efficient than baseline methods.

## 1 Introduction

Clustering is an important approach to data mining and knowledge discovery. Particularly, spectral clustering [Weiss, 1999; Shi and Malik, 2000; Ng *et al.*, 2002; Von Luxburg, 2007] has superior performance than k-means clustering [Steinhaus and others, 1956], hierarchical clustering [Johnson, 1967], DBSCAN [Ester *et al.*, 1996], and mixtures of probabilistic principal component analyzers [Tipping and Bishop, 1999] in many applications. Roughly speaking, spectral clustering consists of two steps: 1) construct an affinity matrix in which each element denotes the similarity between two data points; 2) perform normalized cut [Shi and Malik, 2000] on the graph corresponding to the affinity matrix. K-nearest neighbors (K-NN) and Gaussian kernel $k(\boldsymbol{x}, \boldsymbol{y}) = \exp(-\|\boldsymbol{x} - \boldsymbol{y}\|^2/(2\varsigma^2))$ are two popular methods to construct affinity matrices, where $k$ and $\varsigma$ are hyperparameters.

As the performance of spectral clustering heavily relies on the quality of affinity matrix, in recent years, a variety of methods have been proposed to construct or learn affinity matrices for spectral clustering. Many of them are in the framework of self-expressive [Roweis and Saul, 2000; Elhamifar and Vidal, 2013] model, i.e., minimize$_C \frac{1}{2}\|\boldsymbol{X} - \boldsymbol{X}\boldsymbol{C}\|_F^2 + \lambda\mathcal{R}(\boldsymbol{C})$. Here the columns of $\boldsymbol{X} \in \mathbb{R}^{m \times n}$ are the data points drawn from a union of subspaces. $\boldsymbol{C} \in \mathbb{R}^{n \times n}$ is a coefficient matrix. $\mathcal{R}(\boldsymbol{C})$ denotes a regularization operator on $\boldsymbol{C}$. $\lambda$ is a hyperparameter to be determined in advance.

---

*Corresponding author

Elhamifar and Vidal [2013] proposed to use $\mathcal{R}(\boldsymbol{C}) = \|\boldsymbol{C}\|_1 := \sum_{i=1}^{n} \sum_{j=1}^{n} |c_{ij}|$ under a constraint $\mathrm{diag}(\boldsymbol{C}) = \boldsymbol{0}$. In [Elhamifar and Vidal, 2013], the affinity matrix for spectral clustering is given by $\boldsymbol{A} = |\boldsymbol{C}| + |\boldsymbol{C}|^{\top}$. The method is called Sparse Subspace Clustering (SSC). Some theoretical results of SSC can be found in [Wang and Xu, 2013; Soltanolkotabi *et al.*, 2014].

Following the self-expressive framework, Liu *et al.* [2013] let $\mathcal{R}(\boldsymbol{C}) = \|\boldsymbol{C}\|_*$ (nuclear norm of $\boldsymbol{C}$) and proposed a Low-Rank Representation (LRR) method for subspace clustering. Lu *et al.* [2012] and Pan Ji *et al.* [2014] used the least squares representation (LSR) model for subspace clustering. A few variants of LRR and SSC can be found in [Patel and Vidal, 2014; Li and Vidal, 2015; Patel *et al.*, 2015; Shen and Li, 2016; Li and Vidal, 2016; Fan and Chow, 2017; Fan *et al.*, 2018; Lu *et al.*, 2018; Pan and Kang, 2021; Kang *et al.*, 2022]. Recently, deep learning methods were also used to learn affinity matrices for spectral clustering [Ji *et al.*, 2017; Zhang *et al.*, 2019b,a; Lv *et al.*, 2021] and have achieved state-of-the-art performance on many benchmark datasets.

One common limitation of these spectral or subspace clustering methods is that they have at least one hyperparameter to determine. In the codes of SSC[2] and its variants provided by their authors, there is usually one more thresholding parameter for affinity matrix, which affects the clustering accuracy a lot. In the deep learning clustering methods such as [Ji *et al.*, 2017] and [Zhang *et al.*, 2019b], we need to determine the network structures and regularization parameters, which is much more difficult. Since clustering is an unsupervised learning problem, the hyperparameters cannot be tuned by cross-validation widely used in supervised learning. Thus we have to tune the hyperparameters in spectral clustering by experience, which is difficult when the dataset is quite different from those in our experience and/or the inter-class similarity is high compared to the intra-cluster similarity. Note that SSC, LRR, and their kernel or deep learning extensions have quadratic or even cubic time complexity (per iteration), which further increases the difficulty of hyperparameter selection in clustering large datasets, though there have been a few works improving the computational efficiency [Peng *et al.*, 2013; Cai and Chen, 2014; Wang *et al.*, 2014; Peng *et al.*, 2015; You *et al.*, 2016a,b; Li and Zhao, 2017; You *et al.*, 2018; Matsushima and Brbic, 2019; Li *et al.*, 2020; Chen *et al.*, 2020; Kang *et al.*, 2020; Fan, 2021; Cai *et al.*, 2022]. On the other hand, different datasets often require different AMC methods, which is hard to tackle by experience.

This paper aims at model and hyperparameter selection for spectral clustering and wants to improve the convenience, accuracy, and efficiency of spectral clustering. Our contributions are as follows.

- We propose a *relative-eigen-gap* based automated spectral clustering (AutoSC) method. It finds the Laplacian matrix with largest *relative-eigen-gap* among a set of candidates constructed by different models with different hyperparameters.

- We also implement the AutoSC method via Bayesian optimization. The method can select the possibly best model and optimize the hyperparameters automatically. Note that any AMC methods (e.g. SSC) can be included in the framework of AutoSC.

- To improve the accuracy and efficiency of AutoSC, we propose a new AMC method based on least squares representation and thresholding and prove its effectiveness theoretically.

- We provide an extension for AutoSC to cluster large-scale datasets.

Experiments on seven benchmark image datasets demonstrate the effectiveness of our method. Particularly, our method outperforms state-of-the-art methods of large-scale clustering.

## 2 Related work

**Exploiting eigenvalue information for clustering** As the number of zero eigenvalues of a Laplacian matrix is equal to the number of connected components of the graph [Von Luxburg, 2007], a few researchers took advantage of eigenvalue information in spectral clustering [Meila *et al.*, 2005; Meila and Shortreed, 2006; Ji *et al.*, 2015; Hu *et al.*, 2017; Lu *et al.*, 2018]. For instance, Ji *et al.* [2015] utilized eigen-gap to determine the rank of the Shape Interaction Matrix. But the method requires determining another hyperparameter $\gamma$ beforehand and needs to perform spectral clustering multiple times. The methods of [Meila *et al.*, 2005; Hu *et al.*, 2017; Lu *et al.*, 2018] are based on

---

[2]Wang and Xu [2013] and Soltanolkotabi *et al.* [2014] provided lower and upper bounds for the $\lambda$ in SSC theoretically, which however depend on the unknown noise level.

iterative optimization (need to perform eigenvalue decomposition at every iteration) and hence are not effective in handling large-scale datasets. In addition, the BDR method of [Lu *et al.*, 2018] has two hyperparameters $(\lambda, \gamma)$ to determine by experience, although it outperformed SSC and LRR on some datasets. A comparison is shown in Figure 1.

**Automated machine learning**   Automated model and hyperparameter selection for supervised learning have been extensively studied [Hutter *et al.*, 2019]. In contrast, the study for unsupervised learning is very limited. The reason is that in unsupervised learning there is no ground truth or reliable metric to evaluate the performance of algorithms. Concurrently to our work, Poulakis [2020] also attempted to do automated clustering. Specifically, Poulakis [2020] proposed to use meta-learning to select clustering algorithm and use a heuristic combination of some clustering validity metrics such as Silhouette coefficient [Liu *et al.*, 2010] and S_Dbw [Halkidi and Vazirgiannis, 2001] as an objective to maximize via grid search or Bayesian optimization [Jones *et al.*, 1998]. One problem is that these metrics are mainly based on Euclidean distance or densities and hence may not be suitable to evaluate the clustering performance of non-distance or non-density based clustering algorithms. Another one is that there is no unified metric to compare different clustering algorithms.

# 3   Automated Spectral Clustering (AutoSC)

## 3.1   Preliminary Knowledge

Let $A \in \mathbb{R}^{n \times n}$ be an affinity matrix constructed from a given data matrix $X \in \mathbb{R}^{m \times n}$. The corresponding graph is denoted by $G = (V, E)$, where $V = \{v_1, \ldots, v_n\}$ is the vertex set and $E = \{e_1, \ldots, e_l\}$ is the edge set. The degree matrix of a graph $G$ is defined as $D = \text{diag}(A\mathbf{1})$, where $\mathbf{1} = [1, \ldots, 1]^\top$. Our goal is to partition the vertices into $k$ disjoint nonempty subsets $C_1, \ldots, C_k$. Let $\mathcal{C} = \{C_1, \ldots, C_k\}$. It is expected to find a partition $\mathcal{C}$ that minimizes the following metric.

**Definition 3.1** (MNCut). The multiway normalized cut (MNCut) [Meila, 2001] is defined as

$$\text{MNCut}(\mathcal{C}) = \sum_{i=1}^{k} \sum_{j \neq i} \frac{\text{Cut}(C_i, C_j)}{\text{Vol}(C_i)}, \tag{1}$$

where $\text{Cut}(C_i, C_j) = \sum_{u \in C_i} \sum_{v \in C_j} A_{uv}$ and $\text{Vol}(C_i)$ denotes the sum of vertex degrees of $C_i$.

The normalized graph Laplacian matrix is defined as

$$L = I - D^{-1/2} A D^{-1/2}, \tag{2}$$

where $I$ is an identity matrix. The normalized graph Laplacian is often more effective than the unnormalized one in spectral clustering (some theoretical justification was given by [Von Luxburg, 2007]). Let $\sigma_i(L)$ be the $i$-th smallest eigenvalue of $L$. The following claim shows the connection between $\text{MNCut}(\mathcal{C})$ and $L$.

**Claim 3.2.** *The sum of the $k$ smallest singular values of $L$ quantifies the potential connectivity among $C_1, \ldots, C_k$:* $\text{MNCut}(\mathcal{C}) \geq \sum_{i=1}^{k} \sigma_i(L)$.

The claim can be easily proved by using Lemma 4 of [Meila, 2001]. We defer all proof of this paper to the appendices. Because the multiplicity $k$ of the eigenvalue 0 of $L$ equals the number of connected components in $G$ [Von Luxburg, 2007], we expect to construct an affinity matrix $A$ from $X$ such that $L$ has $k$ zero eigenvalues. Thus the optimal partition means $\text{MNCut}(\mathcal{C}) = \sum_{i=1}^{k} \sigma_i(L) = 0$.

## 3.2   Relative Eigen-Gap Guided Search

In practice, we may construct an $A$ such that $\sum_{i=1}^{k} \sigma_i(L)$ is as small as possible because guaranteeing zero eigenvalues is difficult. But this is not enough because $L$ may have $k + 1$ or more very small or even zero eigenvalues. The second smallest eigenvalue of the Laplacian matrix of a graph $G$ is called the algebraic connectivity of G (denoted by $ac(G)$) [Fiedler, 1973]. We have $ac(G) = 0$ if and only if $G$ is not connected. When $G$ has $k$ disjointed components, there are $k$ algebraic connectivities, denoted by $ac(C_1), \ldots, ac(C_k)$. Based on this, we have

**Claim 3.3.** *The $k+1^{th}$ smallest eigenvalue of $\boldsymbol{L}$ quantifies the least potential connectivity of partitions $C_1, \ldots, C_k$ of $\mathcal{C}$:*

$$\min_{1 \leq i \leq k} \mathrm{MNCut}(C_i) \geq \sigma_{k+1}(\boldsymbol{L}). \tag{3}$$

In other words, $\sigma_{k+1}(\boldsymbol{L})$ measures the difficulty in segmenting each of $C_i$ into two subsets. Hence, when $\sigma_{k+1}(\boldsymbol{L})$ is large, the partitions $C_1, \ldots, C_k$ are stable. Based on Claim 3.2 and Claim 3.3, we may construct an $\boldsymbol{A}$ that has small $\sum_{i=1}^{k} \sigma_i(\boldsymbol{L})$ and large $\sigma_{k+1}(\boldsymbol{L})$ simultaneously, by solving

$$\underset{\theta}{\mathrm{maximize}} \ \ \sigma_{k+1}(\boldsymbol{L}) - \frac{1}{k} \sum_{i=1}^{k} \sigma_i(\boldsymbol{L}),$$

$$\text{subject to } \boldsymbol{L} = \boldsymbol{I} - \boldsymbol{D}^{-1/2} \boldsymbol{A} \boldsymbol{D}^{-1/2}, \ \boldsymbol{A} = f_\theta(\boldsymbol{X}). \tag{4}$$

where $f_\theta : \mathbb{R}^{m \times n} \to \mathbb{R}^{n \times n}$ is a function with parameter $\theta$, e.g. $\boldsymbol{A} = [\exp(-\|\boldsymbol{x}_i - \boldsymbol{x}_j\|^2 / (2\varsigma^2))] - \boldsymbol{I}$. It is difficult to solve (4) because of the composition of $f_\theta$, symmetric normalized Laplacian, and eigenvalue decomposition. On the other hand, in (4), we have to choose $f$ in advance, which requires domain expertise or strong experience because different dataset usually needs different $f$.

Note that different $f$ can result in very different distributions of eigenvalues and the small eigenvalues are sensitive to $f$, $\theta$, and noise. Hence the objective in (4) is not effective to compared different $f$ and $\theta$. In this paper, we define a new metric *relative-eigen-gap* as follows

$$\mathrm{reg}(\boldsymbol{L}) := \frac{\sigma_{k+1}(\boldsymbol{L}) - \frac{1}{k} \sum_{i=1}^{k} \sigma_i(\boldsymbol{L})}{\frac{1}{k} \sum_{i=1}^{k} \sigma_i(\boldsymbol{L}) + \varepsilon}, \tag{5}$$

where $\varepsilon$ is a small constant (e.g. $10^{-6}$) to avoid zero denominator. $\mathrm{reg}(\boldsymbol{L})$ is not sensitive to the scale of the small eigenvalues. Therefore, instead of (4), we propose to solve

$$\underset{(f,\theta) \in \mathcal{F} \times \Theta}{\mathrm{maximize}} \ \ \mathrm{reg}(\boldsymbol{L}),$$

$$\text{subject to } \boldsymbol{L} = \boldsymbol{I} - \boldsymbol{D}^{-1/2} \boldsymbol{A} \boldsymbol{D}^{-1/2}, \ \boldsymbol{A} = f_\theta(\boldsymbol{X}), \tag{6}$$

where $\mathcal{F}$ is a set of pre-defined functions and $\Theta$ is a set of hyperparameters. In fact, (6) is equivalent to choosing one $\boldsymbol{A}$ (or $\boldsymbol{L}$) from a set of candidates constructed by different $f$ with different $\theta$, of which the *relative-eigen-gap* is largest. The best $\theta$ can be found using grid search (or even random search). For convenience, we call the method AutoSC-GD. Table 1 shows a few examples of $f$ and its parameters. One may use a weighted sum of affinity matrices given by different $f$, like [Huang *et al.*, 2012], which however will introduce more hyperparameters.

Table 1: A few examples of $f$ and its $\theta$ for AMC (AASC: [Huang *et al.*, 2012])

| $f$ | K-NN | $\epsilon$-neighborhood | Gaussian kernel | SSC | LRR | LSR | KSSC | AASC |
|---|---|---|---|---|---|---|---|---|
| $\theta$ | $K$ | $\epsilon$ | $\sigma$ | $\lambda$ | $\lambda$ | $\lambda$ | $\lambda, \sigma$ | $\sigma_1, \sigma_2, \ldots$ |

The following theorem[3] shows the connection between $\mathrm{reg}(\boldsymbol{L})$ and the stability of the clustering $\mathcal{C}$.

**Theorem 3.4.** *Let $\mathcal{C}$ and $\mathcal{C}'$ be two partitions of the vertices of $G$, where $|\mathcal{C}| = |\mathcal{C}'| = k$. Define the distance between $\mathcal{C}$ and $\mathcal{C}'$ as $\mathrm{dist}(\mathcal{C}, \mathcal{C}') = 1 - \frac{1}{k} \sum_{C_i \in \mathcal{C}} \sum_{C'_j \in \mathcal{C}'} \frac{(\mathrm{Vol}(C_i \cap C'_j))^2}{\mathrm{Vol}(C_i) \mathrm{Vol}(C'_j)}$. Suppose $\eta k \varepsilon \geq \sum_{i=1}^{k} \sigma_i(\boldsymbol{L}) \geq k\varepsilon$ and $\mathrm{reg}(\boldsymbol{L}) > (k-1)\eta/2$. Let $\delta = \max\big(\mathrm{MNCut}(\mathcal{C}) - \sum_{i=1}^{k} \sigma_i(\boldsymbol{L}), \mathrm{MNCut}(\mathcal{C}') - \sum_{i=1}^{k} \sigma_i(\boldsymbol{L})\big)$. Then*

$$\mathrm{dist}(\mathcal{C}, \mathcal{C}') < \frac{1.5\delta \varepsilon^{-1}}{\mathrm{reg}(\boldsymbol{L}) + (1-k)\eta/2}. \tag{7}$$

It indicates that when $\mathrm{reg}(\boldsymbol{L})$ is large and $\delta$ is small, the partitions $\mathcal{C}$ and $\mathcal{C}'$ are close to each other. Thus, the clustering has high stability. When $\sum_{i=1}^{k} \sigma_i(\boldsymbol{L}) = k\varepsilon$, we have $\mathrm{dist}(\mathcal{C}, \mathcal{C}') < \frac{6\delta}{\sigma_{k+1}(\boldsymbol{L}) - k\varepsilon}$, which means the larger $\sigma_{k+1}(\boldsymbol{L})$ the more stable clustering.

---

[3]This theorem is a modified version of Theorem 1 in [Meila *et al.*, 2005], which is for the eigen-gap $\sigma_{k+1} - \sigma_k$ of $\boldsymbol{L}$. Here we consider $\mathrm{reg}(\boldsymbol{L})$ instead.

**Compare reg($L$) with [Meila and Shortreed, 2006]** It is worth noting that Meila and Shortreed [2006] proposed to minimize $\mathrm{J}(L) := \frac{1}{k} \sum_{i=1}^{k} \sigma_i(L) + \alpha \left(\sigma_k(L) - \sigma_{k+1}(L)\right)^2$ to find a good affinity matrix for spectral clustering. Although reg($L$) and $\mathrm{J}(L)$ seem similar, they are essentially different. As mentioned before, different AMC methods may lead to different scales for the small eigenvalues, which makes it difficult to compare different AMC methods using $\mathrm{J}(L)$. In addition, $\mathrm{J}(L)$ has a hyperparameter $\alpha$ to determine beforehand, which violates our goal of searching models and hyperparameters. A comparative study ($\alpha \leq 0$) is in Section 4.1 (Table 3).

### 3.3 AutoSC via Bayesian Optimization

Bayesian optimization (BO) [Jones *et al.*, 1998] has become a promising tool for hyperparameter optimization of supervised machine learning algorithms [Snoek *et al.*, 2012; Klein *et al.*, 2017]. Given a black-box function $g : \mathcal{X} \to \mathbb{R}$, BO aims to find an $x^* \in \mathcal{X}$ that globally minimizes $g$ and usually has three steps. The first step is finding the most promising point $x_{t+1} \in \operatorname{argmax}_x a_{p(g)}(x)$ by numerical optimization, where $a_{p(g)} : \mathcal{X} \to \mathbb{R}$ is an acquisition function (e.g. Expected Improvement) relying on an prior $p(g)$ (e.g. Gaussian processes [Williams and Rasmussen, 2006]). The second step is evaluating the expensive and possibly noisy function $y_{t+1} \sim g(x) + \mathcal{N}(0, \sigma^2)$ and adding the new sample $(x_{t+1}, y_{t+1})$ to the observation set $\mathcal{D}_t = \{(x_1, y_1), \ldots, (x_t, y_t)\}$. The last step is updating $p(g)$ and $a_{p(g)}$ using $\mathcal{D}_{t+1}$.

As an alternative to the grid search for (6), we can maximize reg($L$) via BO. Suppose we have a set of different AMC models, i.e., $\mathcal{F} = \{f_1, f_2, \ldots, f_M\}$. For $i = 1, 2, \ldots, M$, let

$$g_i(\theta^{(i)}) := -\mathrm{reg}(L(f_i(\theta^{(i)}|\mathbb{X}))),$$

where $\theta^{(i)}$ denotes the hyperparameters in $f_i$ and $\mathbb{X}$ denotes the dataset. Then we use BO to find

$$\theta_*^{(i)} = \operatorname*{argmin}_{\theta^{(i)} \in S^{(i)}} g_i(\theta^{(i)}), \tag{8}$$

where $S^{(i)}$ denotes the set of constraints. Finally we get the best model with its best hyperparameters

$$f_\star(\theta_*^{(\star)}|\mathbb{X}), \quad \text{where } \star = \operatorname*{argmin}_{1 \leq i \leq M} g_i(\theta_*^{(i)}). \tag{9}$$

For convenience, we denote the method by AutoSC-BO. Note that $\mathcal{F}$ can include any AMC methods such as those in Table 1 and even DSC [Ji *et al.*, 2017] (see Appendix D.5).

In AutoSC-BO, we use Expected Improvement (EI) acquisition function

$$a_{\mathrm{EI}}(s|\mathcal{D}_t) = \mathbb{E}_p\left[\max(g_{\min} - g(s), 0)\right], \tag{10}$$

where $g_{\min}$ is the best function value known. The closed-form formulation is

$$a_{\mathrm{EI}}(s|\mathcal{D}_t) = (g_{\min} - \mu)\Phi\left(\frac{g_{\min} - \mu}{\sigma}\right) + \phi\left(\frac{g_{\min} - \mu}{\sigma}\right), \tag{11}$$

where $\mu = \mu(s|\mathcal{D}_t, \theta_K)$ and $\sigma = \sigma(s|\mathcal{D}_t, \theta_K)$ are the mean value and variance of the Gaussian process, $\phi$ and $\Phi$ are standard Gaussian cumulative density function and probability density function respectively, and $\theta_K$ denotes the hyperparameters of the Gaussian process. For the covariance function, we use the automatic relevance determination (ARD) Matérn $5/2$ kernel [Matérn, 2013]

$$k_{M52}(s, s') = \theta_0 \left(1 + \sqrt{5r^2(s, s')} + \frac{5}{3}r^2(s, s')\right) \times \exp\left(-\sqrt{5r^2(s, s')}\right), \tag{12}$$

where $r^2(s, s') = \sum_{j=1}^{d}(s_j - s_j')^2/\theta_j^2$.

### 3.4 Discussion on AMC Methods for AutoSC and LSR with Thresholding

In AutoSC, the size of searching space is $|\mathcal{F}| \times \prod_j |\theta_j|$, which should be large enough to include effective models and their hyperparameters. A large number of works have shown that the self-expressive models [Elhamifar and Vidal, 2013; Liu *et al.*, 2013; Lu *et al.*, 2018; Ji *et al.*, 2017] often outperform other AMC models such as Gaussian kernel. However, the self-expressive model based

AMC methods often require iterative optimization and has at least quadratic time complexity per iteration, which leads to huge time cost in AutoSC. Although LSR [Lu *et al.*, 2012] has closed-form solution, the clustering accuracy is not satisfactory [Lu *et al.*, 2018]. In this work, we will show that LSR with a simple post-processing operation can be a good AMC method and can outperform SSC, LRR, and BDR [Lu *et al.*, 2018]. Specifically, the LSR model is given as

$$\underset{C}{\text{minimize}} \quad \frac{1}{2}\|X - XC\|_F^2 + \frac{\lambda}{2}\|C\|_F^2, \tag{13}$$

of which the closed-form solution is $C = (X^\top X + \lambda I)^{-1} X^\top X$. Let diag($C$) = $\mathbf{0}$ and $C \leftarrow |C|$, the affinity matrix can be constructed as $A = (C + C^\top)/2$. One problem is that the off-diagonal elements of $A$ are dense (leading to a connected graph), which can result in low clustering accuracy. Therefore, we propose to truncate $C$ by keeping only the largest $\tau$ elements of each column of $C$. Nevertheless, it is not easy to determine $\tau$ beforehand. When $\tau$ is too small, the corresponding graph will have $k+1$ or more connected components. When $\tau$ is too large, the corresponding graph will have $k-1$ or less connected components. However, $\tau$ can be automatically determined by our AutoSC-GD and AutoSC-BO.

In the case that the data have some low-dimensional nonlinear structures, the similarity between pair-wise columns of $X$ cannot be well recognized by the linear regression (13). Therefore, we also consider the following nonlinear regression model

$$\underset{C}{\text{minimize}} \quad \frac{1}{2}\|\phi(X) - \phi(X)C\|_F^2 + \frac{\lambda}{2}\|C\|_F^2, \tag{14}$$

where $\phi$ denotes a nonlinear feature map performed on each column of the matrix, i.e. $\phi(X) = [\phi(x_1), \dots, \phi(x_n)]$. In (14), letting $\phi$ be some feature map induced by a kernel function $k(\cdot, \cdot)$ (e.g. polynomial kernel $k(x_i, x_j) = (x_i^\top x_j + b)^q$ and Gaussian kernel), we get the kernel LSR (KLSR):

$$\underset{C}{\text{minimize}} \quad \frac{1}{2}\text{Tr}\left(K - 2KC + C^\top KC\right) + \frac{\lambda}{2}\|C\|_F^2, \tag{15}$$

where $K = \phi(X)^\top \phi(X)$ and $[K]_{ij} = k(x_i, x_j)$. The closed-form solution is $C = (K+\lambda I)^{-1}K$. The post-processing is the same as that for the solution of LSR.

We introduce the following property, a necessary condition of successful subspace clustering, which is similar to the one used in [Wang and Xu, 2013; Soltanolkotabi *et al.*, 2014].

**Definition 3.5** (Subspace Detection Property). A symmetric affinity matrix $A$ obtained from $X$ has subspace detection property if for all $i$, the nonzero elements of $a_i$ correspond to the columns of $X$ in the same subspace as $x_i$.

For convenience, let $\pi(i)$ be the index of the subspace $x_i$ belongs to and $C_j$ be the index set of the columns of $X$ in subspace $j$. We consider the following deterministic model.

**Definition 3.6** (Deterministic Model). The columns of $X \in \mathbb{R}^{m \times n}$ are drawn from a union of $k$ different subspaces and are further corrupted by noise, where $\dim(\mathcal{S}_1 \cup \cdots \cup \mathcal{S}_k) = d < m \leq n$. Let $X = U\Sigma V^\top$ be the SVD of $X$, where $\Sigma = \text{diag}(\sigma_1, \dots, \sigma_n)$ and $\sigma_1 \geq \sigma_2 \geq \cdots \sigma_n$. Let $\gamma = \sigma_{d+1}/\sigma_d$. Denote $v_i = (v_{i1}, \dots, v_{in})$ the $i$-th row of $V$ and let $\bar{v}_i = (v_{i1}, \dots, v_{id})$. Suppose the following conditions hold[4]: 1) for every $i \in [n]$, the $\bar{\tau}$-th largest element of $\{|\bar{v}_i^\top \bar{v}_j| : j \in C_{\pi(i)}\}$ is greater than $\alpha$; 2) $\max_{i \in [n]} \max_{j \in [n] \setminus C_{\pi(i)}} |\bar{v}_i^\top \bar{v}_j| \leq \beta$; 3) $\max_{i,j,l} |v_{il}v_{jl}| \leq \mu$.

Then the following theorem verifies the effectiveness of (13) followed by the truncation (thresholding) operation in subspace detection.

**Theorem 3.7.** *Suppose $X$ is given by Definition 3.6 and $C$ is given by* (13) *with*

$$\frac{\left(\rho - \sqrt{\rho^2 - 4(2\mu d - \Delta)(2\mu m - 2\mu d - \Delta)}\right)\sigma_d^2}{4\mu d - 2\Delta} < \lambda < \frac{\left(\rho + \sqrt{\rho^2 - 4(2\mu d - \Delta)(2\mu m - 2\mu d - \Delta)}\right)\sigma_d^2}{4\mu d - 2\Delta} \tag{16}$$

*where $\rho = 2\mu m\gamma^2 - \Delta(1 + \gamma^2)$ and $\Delta = \alpha - \beta$. Then the $C$ truncated by $\tau \leq \bar{\tau}$ has the subspace detection property.*

---

[4] $\gamma$ measures the noise level, $\beta$ is dominated by the difference between subspaces, and $\mu$ quantifies the incoherence in the singular vectors.

In Theorem 3.7, the width of the range of $\lambda$ is $w = \frac{\sqrt{\rho^2 - 4(2\mu d - \Delta)(2\mu m - 2\mu d - \Delta)}\sigma_d^2}{2\mu d - \Delta}$. We see that a larger $\sigma_d$, $\Delta$, or smaller $\gamma$, $d$ leads to a wider range of $\lambda$, which corresponds to a simper clustering problem. When $\rho^2 \leq 4(2\mu d - \Delta)(2\mu m - 2\mu d - \Delta)$, $\lambda$ does not exist. Theorem 3.7 can be extended to the kernel case (15) without the restriction of $d < m$ even when the columns of $\boldsymbol{X}$ are drawn from a union of nonlinear low-dimensional manifolds. See Definition C.1, Definition C.2, and Theorem C.3 in Appendix C. Based on Theorem 3.7 and Theorem C.3, the follow proposition indicates that AutoSC can cluster the data correctly.

**Proposition 3.8.** *Suppose the affinity matrix $\boldsymbol{A}$ given by AutoSC has the subspace or manifold detection property (defined in Appendix C) and $\mathrm{reg}(\boldsymbol{L}) = \frac{\sigma_{k+1}}{\epsilon} > 0$. Then each component of G consists of all columns of $\boldsymbol{X}$ in the same subspace or manifold.*

Now we see that LSR and KLSR with thresholding can provide effective self-expressive affinity matrices for AutoSC without performing iterative optimization. On the other hand, the relative-eigen-gap is able to compare LSR with KLSR, compare different kernels, and evaluate $\lambda$, $\tau$, and kernel parameters. Note that if we use SSC and KSSC instead of LSR and KLSR, AutoSC will be very time-consuming. If we use LSR and KLSR without thresholding, AutoSC may not provide high clustering accuracy. We hope that AutoSC is not only automatic but also accurate and efficient.

### 3.5 AutoSC+NSE for Large-Scale Data

Since the time and space complexity of AutoSC are quadratic with $n$, it cannot be directly applied to large-scale datasets. To solve the problem, we propose to perform Algorithm 1 on a set of $s$ landmarks of the data (denoted by $\hat{\boldsymbol{X}}$) to get a $\hat{\boldsymbol{Z}}$. The landmarks can be generated by k-means or randomly. Then we regard $\hat{\boldsymbol{Z}}$ as a feature matrix and learn a map $g : \mathbb{R}^m \rightarrow \mathbb{R}^k$ from $\hat{\boldsymbol{X}}$ to $\hat{\boldsymbol{Z}}$. According to the universal approximation theorem [Sonoda and Murata, 2017] of neural networks, we approximate $g$ by a two-layer neural network and solve

$$\underset{\boldsymbol{W}_1, \boldsymbol{W}_2, \boldsymbol{b}_1, \boldsymbol{b}_2}{\text{minimize}} \quad \frac{1}{2s} \|\hat{\boldsymbol{Z}} - \boldsymbol{W}_2 \mathrm{ReLU}(\boldsymbol{W}_1 \hat{\boldsymbol{X}} + \boldsymbol{b}_1 \boldsymbol{1}_s^\top) - \boldsymbol{b}_2 \boldsymbol{1}_s^\top\|_F^2 + \frac{\gamma}{2} \left( \|\boldsymbol{W}_1\|_F^2 + \|\boldsymbol{W}_2\|_F^2 \right), \quad (17)$$

where $\boldsymbol{W}_1 \in \mathbb{R}^{d \times m}$, $\boldsymbol{W}_2 \in \mathbb{R}^{k \times d}$, $\boldsymbol{b}_1 \in \mathbb{R}^d$, and $\boldsymbol{b}_2 \in \mathbb{R}^k$. Since $\boldsymbol{A}$ is sparse, $k$ is often less than $m$, and a neural network is used, we call (17) Neural Sparse Embedding (NSE). We use mini-batch Adam [Kingma and Ba, 2014] to solve NSE. It is worth noting that NSE is different the method proposed by [Li *et al.*, 2020]. In [Li *et al.*, 2020], the regression is for an affinity matrix, which leads to high computational cost. The network learned from (17) is applied to $\boldsymbol{X}$ to extract a $k$-dimensional feature matrix $\boldsymbol{Z}$:

$$\boldsymbol{Z} = \hat{g}(\boldsymbol{X}) = \boldsymbol{W}_2 \mathrm{ReLU}(\boldsymbol{W}_1 \boldsymbol{X} + \boldsymbol{b}_1 \boldsymbol{1}_n^\top) + \boldsymbol{b}_2 \boldsymbol{1}_n^\top. \quad (18)$$

Finally, we perform k-means on $\boldsymbol{Z}$ to get the clusters. The procedures are summarized into Algorithm 2 (see Appendix B). Note that the time complexity of AutoSC+NSE is $O(dmn + \tilde{d}s^2)$, where $\tilde{d}$ depends on the specific AMC method. When $s \ll n$, the time complexity of AutoSC+NSE is linear with the number of data points $n$. Proposition C.4 in Appendix C.2 shows that a small number of hidden nodes in NSE are sufficient to make the clustering succeed.

## 4 Experiments

We test our AutoSC on Extended Yale B Face [Kuang-Chih *et al.*, 2005], ORL Face [Samaria and Harter, 1994], COIL20 [Nene *et al.*, 1996], AR Face [Martínez and Kak, 2001], MNIST [LeCun *et al.*, 1998], Fashion-MNIST [Xiao *et al.*, 2017], GTSRB [Stallkamp *et al.*, 2012], subsets and extracted features of MNIST and Fashion-MNIST. The descriptions for the datasets are in Appendix D.1. Our MATLAB codes are available at `https://github.com/jicongfan/Automated-Spectral-Clustering`.

### 4.1 Intuitive validation of AutoSC

**Performance of Relative-Eigen-Gap** First, we use LSR and KLSR to show the effectiveness of the proposed $\mathrm{reg}(\boldsymbol{L})$. Figure 1(i) shows an intuitive example of the performance of LSR and KLSR

in clustering a subset of the Extended Yale B database, where for (15) we use Gaussian kernel with $\varsigma = \frac{1}{n^2} \sum_{ij} \|\boldsymbol{x}_i - \boldsymbol{x}_j\|$. We see that: 1) in LSR and KLSR, for a fixed $\lambda$ (or $\tau$), the $\tau$ (or $\lambda$) with larger reg($\boldsymbol{L}$) provides higher clustering accuracy; 2) for a fixed $\lambda$ and a fixed $\tau$, if LSR has a larger reg($\boldsymbol{L}$), its clustering accuracy is higher than that of KLSR, and vice versa. We conclude that roughly a larger reg($\boldsymbol{L}$) indeed leads to a higher clustering accuracy, which is consistent with our theoretical analysis in Section 3.2.

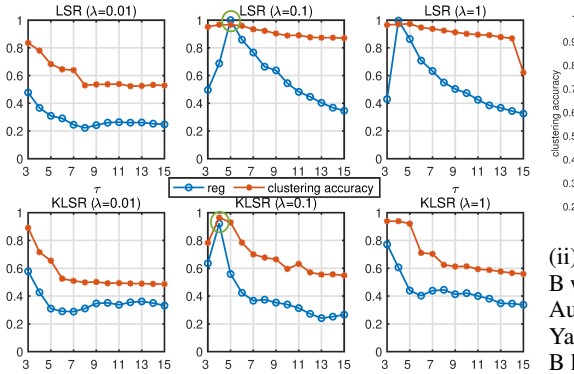
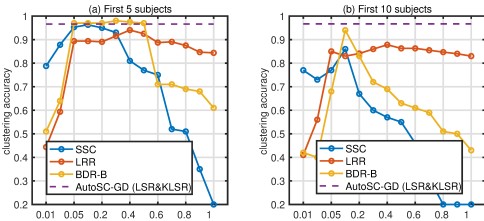

(ii) Clustering accuracies of SSC, LRR, and BDR-B with different hyperparameter $\lambda$ in comparison to AutoSC-GD with LSR and KLSR on the Extended Yale Face B database. The value of $\lambda$ used in BDR-B has been divided by 10. The $\gamma$ in BDR-B is chosen from $\{0.01, 0.1, 1\}$ and the best one is used for each $\lambda$. In Case (b), the time costs of SSC, LRR, BDR-B, and AutoSC-GD are 9.5s, 33.0s, 7.6s, and 1.6s respectively.

(i) Clustering accuracy and reg($\boldsymbol{L}$) (rescaled between 0 and 1) and clustering accuracy of LSR and KLSR on the first 5 subjects of the Extended Yale Face B database.

Figure 1: Examples about reg($\boldsymbol{L}$), clustering accuracy, and hyperparameters on Extended Yale B.

Now we show the superiority of LSR and KLSR compared to a few important AMC methods. Figure 1(ii) shows the clustering accuracy of SSC [Elhamifar and Vidal, 2013], LRR [Liu *et al.*, 2013], and BDR-B [Lu *et al.*, 2018] with different hyperparameters and our method Auto-GD with LSR and KLSR (detailed by Algorithm 1 in the supplementary material) on the Extended Yale Face B subset. SSC and BDR-B are sensitive to the value of $\lambda$, especially for the relatively difficult task, say Figure 1(ii-b). LRR is not sensitive to the value of $\lambda$ but its accuracy is low. LSR and KLSR are more accurate and efficient than other methods.

**The performance of AutoSC-BO** We show the performance of AutoSC-BO with many AMC methods such as SSC [Elhamifar and Vidal, 2013] and LSR. Taking the KLSR model (15) with polynomial kernel as an example, the parameters are $\theta = (\lambda, b, q, \tau)^\top$ and the constraints are given by $S = \{\lambda \in \mathbb{R} : \lambda_{min} \leq \lambda \leq \lambda_{max}; \ b \in \mathbb{R} : b_{min} \leq b \leq b_{max}; \ q \in \mathbb{Z}^+ : q_{min} \leq q \leq q_{max}; \ \tau \in \mathbb{Z}^+ : \tau_{min} \leq \tau \leq \tau_{max}\}$. More details are in Appendix D.5. Shown in Table 2, larger reg corresponds to higher clustering accuracy and KLSR with polynomial kernel (the optimal $q$ is 1) performs best. Figure 2 shows the performance of KSSC [Patel and Vidal, 2014] and KLSR in each iteration of AutoSC-BO.

Table 2: Clustering accuracy of AutoSC-BO with many methods on Yale Face B dataset (first 10 subjects). All hyperparameters of the kernel functions were optimized via Bayesian optimization.

| AMC | $\epsilon$-neigh -borhood | Polynomial kernel | Gaussian kernel | KSSC (Gauss) | KSSC (Poly) | KLSR (Gauss) | KLSR (Poly) |
|---|---|---|---|---|---|---|---|
| reg$_{max}$ | 0.776 | 1.294 | 1.307 | 0.892 | 1.388 | 2.217 | 2.379 |
| Accuracy | 0.325 | 0.389 | 0.393 | 0.584 | 0.859 | 0.963 | 0.966 |

**Relative-Eigen-Gap versus Eigen-Gap** We compare the proposed relative-eigen-gap (reg($\mathbf{L}$)) with eigen-gap (denoted by eg($\mathbf{L}$)), and the regularizer $J(\mathbf{L}, \alpha)$ proposed by [Meila and Shortreed, 2006] with different $\alpha$. Note that in AutoSC, we need to minimize $J(\mathbf{L}, \alpha)$ instead. The clustering results of AutoSC-GD are reported in Table 3 (details about the datasets are in Section 4). We see that our reg($\mathbf{L}$) outperforms eg($\mathbf{L}$) and $J(\mathbf{L}, \alpha)$ in all cases except AR.

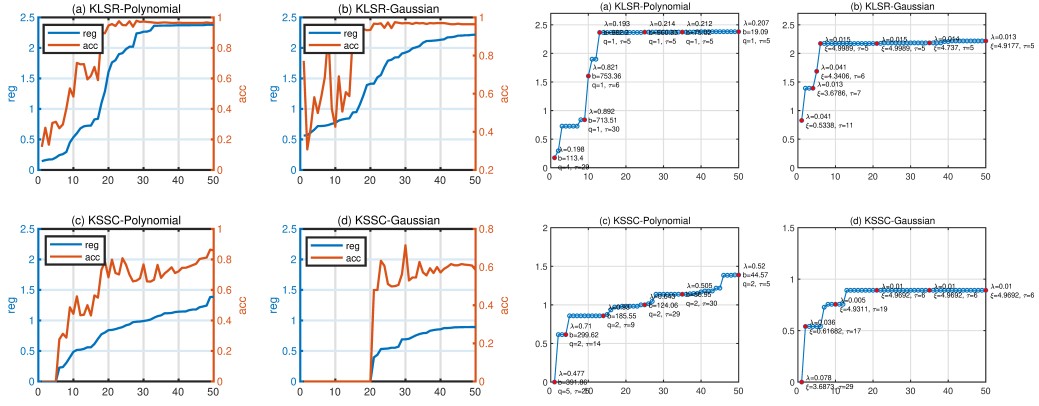

(i) reg($L$) and clustering accuracy of KLSR and KSSC in each iteration of BO.

(ii) reg($L$) and hyperparameters of KLSR and KSSC in each iteration of BO.

Figure 2: AutoSC-BO with KSSC and KLSR on the first 10 subjects of YaleB Face dataset.

Table 3: The comparison of AutoSC-GD with reg($\mathbf{L}$), eg($\mathbf{L}$), and J($\mathbf{L}, \alpha$)

|  | YaleB | ORL | COIL20 | AR | MNIST | F-MNIST |
|---|---|---|---|---|---|---|
| eg($\mathbf{L}$) | 0.790 | 0.768 | 0.619 | 0.805 | 0.663 | 0.562 |
| J($\mathbf{L}, 0$) | 0.823 | **0.795** | 0.750 | 0.804 | 0.726 | 0.516 |
| J($\mathbf{L}, -0.1$) | 0.818 | 0.788 | 0.768 | 0.826 | 0.718 | 0.519 |
| J($\mathbf{L}, -1$) | 0.812 | 0.785 | 0.769 | 0.817 | 0.722 | 0.523 |
| J($\mathbf{L}, -10$) | 0.804 | 0.788 | 0.768 | **0.832** | 0.731 | 0.539 |
| J($\mathbf{L}, -100$) | 0.788 | 0.765 | 0.769 | 0.817 | 0.735 | 0.545 |
| reg($\mathbf{L}$) | **0.897** | **0.795** | **0.782** | 0.786 | **0.755** | **0.595** |

## 4.2 Comparative studies of AutoSC and baselines

First we compare AutoSC with SSC, LRR [Liu *et al.*, 2013], LSR [Lu *et al.*, 2012], EDSC [Pan Ji *et al.*, 2014], KSSC, SSC-OMP [You *et al.*, 2016b], BDR-Z [Lu *et al.*, 2018], and BDR-B [Lu *et al.*, 2018] on six smaller datasets. The clustering accuracy and time cost are reported in Table 4. AutoSC-GD and AutoSC-BO outperformed other methods significantly in almost all cases. SSC-OMP and AutoSC-GD are more efficient than SSC, LRR, EDSC, and KSSC. The time cost of AutoSC-BO is much higher than that of AutoSC-GD because the former optimizes all hyperparameters including $\lambda$, $\tau$, and kernel parameters (e.g. $b$, $q$).

Table 4: Clustering performance on the six small datasets. For the MNIST-1k and Fahsion-MNIST-1k, we report the average results of 20 trials because the subset is formed randomly. AutoSC chose LSR for Yale B and AR and chose KLSR for others datasets. The NMI results are in Table 7 (See Appendix D.3).

|  |  | SSC | LRR | LSR | EDSC | KSSC | SSC-OMP | BDR-Z | BDR-B | AutoSC-GD | AutoSC-BO |
|---|---|---|---|---|---|---|---|---|---|---|---|
| Yale B | acc | 0.723 | 0.643 | 0.592 | 0.806 | 0.649 | 0.768 | 0.596 | 0.719 | **0.897** | **0.909** |
|  | time | 273.8 | 928.1 | 7.3 | 58.6 | 464.3 | **8.9** | 368.8 | 368.8 | 19.1 | 78.3 |
| ORL | acc | 0.711 | 0.762 | 0.680 | 0.712 | 0.707 | 0.665 | 0.739 | 0.735 | **0.795** | **0.803** |
|  | time | 2.7 | 8.8 | **0.5** | 2.0 | 2.6 | **0.4** | 3.9 | 3.9 | 2.3 | 20.5 |
| COIL20 | acc | 0.871 | 0.729 | 0.695 | 0.759 | **0.912** | 0.658 | 0.713 | 0.791 | 0.782 | **0.878** |
|  | time | 61.8 | 221.2 | **1.4** | 15.4 | 100.6 | **2.5** | 86.8 | 86.8 | 7.6 | 39.2 |
| AR | acc | 0.718 | 0.769 | 0.665 | 0.673 | 0.726 | 0.669 | 0.745 | 0.751 | **0.786** | **0.826** |
|  | time | 317.5 | 1220.6 | **14.5** | 69.1 | 627.4 | 57.6 | 578.7 | 578.7 | **43.4** | 130.6 |
| MNIST -1k | acc | 0.596 (0.054) | 0.513 (0.037) | 0.554 (0.041) | 0.536 (0.035) | 0.577 (0.053) | 0.542 (0.038) | 0.576 (0.037) | 0.578 (0.043) | **0.615** (0.041) | **0.619** (0.038) |
|  | time | 24.9 | 69.1 | **0.8** | 5.2 | 32.8 | **1.3** | 26.9 | 26.9 | 2.5 | 21.7 |
| Fashion-MNIST-1k | acc | 0.553 (0.025) | 0.515 (0.014) | 0.563 (0.023) | 0.544 (0.017) | 0.548 (0.016) | 0.566 (0.034) | 0.574 (0.019) | 0.563 (0.031) | **0.581** (0.025) | **0.584** (0.021) |
|  | time | 24.1 | 68.5 | **0.7** | 5.1 | 35.9 | **1.2** | 25.7 | 25.7 | 2.6 | 22.7 |

We compare our method with LSR, LSC-K [Chen and Cai, 2011], SSC-OMP [You *et al.*, 2016b], and S$^5$C [Matsushima and Brbic, 2019], and S$^3$COMP-C [Chen *et al.*, 2020] on the larger datasets. The parameter settings are in Appendix D.6. Table 5 shows the clustering accuracy and standard deviation of 10 repeated trials on the raw-pixel data of MNIST and Fashion-MNIST. Table 6 shows the results on MNIST and Fashion-MNIST features and GTSRB. Our methods have the highest clustering accuracy in every case. Note that if NSE is ablated, the accuracy of AutoSC-GD on MNIST(features)-10k, -20k, and -30k are 0.9772, 0.9783 and 0.9864 respectively, higher than those of AutoSC-GD+NSE, though the time costs increased.

It is worth mentioning that, to the best of our knowledge, the deep clustering method of [Zhang *et al.*, 2019a] has SOTA performance on Yale B (acc=0.98) and ORL (acc=0.89), the method proposed by [Zhang *et al.*, 2019b] has SOTA performance on Fashion-MNIST (acc=0.72), the method proposed by [Mahon and Lukasiewicz, 2021] has SOTA performance on MNIST (acc=0.99), and our method has SOTA performance on GTSRB. Nevertheless, we focus on automated spectral clustering.

Table 5: Clustering accuracy and time cost (second) on MNIST and Fashion MNIST. "—" means the computation is out of memory.

| | | LSR | LSC-K | SSC-OMP | S$^5$C | S$^3$COMP-C | AutoSC-GD+NSE | AutoSC-BO+NSE |
|---|---|---|---|---|---|---|---|---|
| MNIST-10k | acc | 0.583(0.007) | 0.652(0.037) | 0.431(0.014) | 0.646(0.045) | 0.623(0.028) | **0.687**(0.035) | **0.679**(0.034) |
| | time | 154.9 | **18.9** | 26.4 | 82.3 | 710.4/20 | **16.2** | 48.3 |
| MNIST | acc | — | 0.665(0.021) | 0.453(0.017) | 0.627(0.025) | — | **0.755**(0.022) | **0.750**(0.009) |
| | time | — | 329.2 | 1178.3 | 961.5 | — | **86.9** | 123.6 |
| Fashion-MNIST-10k | acc | 0.561(0.008) | 0.571(0.025) | 0.509(0.038) | 0.565(0.021) | 0.569(0.024) | **0.576**(0.011) | **0.572**(0.019) |
| | time | 153.6 | **18.6** | 26.8 | 107.3 | 707.2/20 | **17.3** | 50.9 |
| Fashion-MNIST | acc | — | 0.561(0.015) | 0.359(0.017) | 0.559(0.013) | — | **0.586**(0.008) | **0.578**(0.012) |
| | time | — | 335.1 | 1156.6 | 932.6 | — | **88.7** | 122.8 |

Table 6: Clustering accuracy (mean value and standard deviation) and time cost (second) on MNIST and Fashion-MNIST with feature extraction. "/" means the algorithm was performed on a computational platform not comparable to ours. The underlined values are from [Chen *et al.*, 2020].

| | | LSC-K | SSC-OMP | S$^5$C | S$^3$COMP-C | AutoSC-GD+NSE | AutoSC-BO+NSE |
|---|---|---|---|---|---|---|---|
| MNIST | acc | 0.8659(0.0215) | 0.8159 | 0.7829(0.0283) | 0.9632 | **0.9775**(0.0034) | **0.9741**(0.0044) |
| | time | 273.6 | 280.6 | 907.5 | 416.8 | 59.2 | 115.3 |
| Fashion-MNIST | acc | 0.6131(0.0298) | 0.3796(0.0217) | 0.6057(0.0227) | — | **0.6398**(0.0133) | **0.6461**(0.0104) |
| | time | 251.8 | 1013.9 | 913.2 | — | **61.9** | 112.6 |
| GTSRB | acc | 0.8711(0.0510) | 0.8252 | 0.9044(0.0267) | 0.9554 | **0.9873**(0.0126) | **0.9881**(0.0078) |
| | time | **31.2** | / | 98.7 | / | **16.8** | 69.4 |

## 5  Conclusion

We have proposed an automated spectral clustering method. Extensive experiments showed the effectiveness and superiority of our methods over baseline methods. The efficiency improvement is from the closed-form solutions of the least squares regressions. The accuracy improvement is from the effectiveness of LSR and KLSR with thresholding and the automation of model and hyperparameter selection. One limitation of this work is that we only considered automated spectral clustering while there are many other clustering methods (e.g. [Fan, 2021]) not relying on affinity matrix.

## Acknowledgments

The work of Jicong Fan was supported in part by the Youth program 62106211 of the National Natural Science Foundation of China and the research funding T00120210002 of Shenzhen Research Institute of Big Data. The work of Yiheng Tu was supported in part by the National Natural Science Foundation of China under Grant no. 32171078. The work of Zhao Zhang was supported in part by the National Natural Science Foundation of China under Grant no. 62072151 and Anhui Provincial Natural Science Fund for the Distinguished Young Scholars (2008085J30). The work of Mingbo Zhao was supported in part by the National Natural Science Foundation of China under Grant no. 61971121. The work of Haijun Zhang was supported in part by the National Natural Science Foundation of China under Grant no. 61972112 and no. 61832004, the Guangdong Basic and Applied Basic Research Foundation under Grant no. 2021B1515020088.

The authors appreciate the reviewers' comments and time.

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
