# A Simple Approach to Automated Spectral Clustering Appendices

**Jicong Fan**[1,2], **Yiheng Tu**[3,4], **Zhao Zhang**[5]*, **Mingbo Zhao**[6], **Haijun Zhang**[7]

[1]The Chinese University of Hong Kong, Shenzhen   [2]Shenzhen Research Institute of Big Data
[3]Chinese Academy of Science, Beijing   [4]University of Chinese Academy of Sciences, Beijing
[5]Hefei University of Technology, Hefei   [6]Donghua University, Shanghai
[7]Harbin Institute of Technology, Shenzhen
fanjicong@cuhk.edu.cn   yihengtu@gmail.com   cszzhang@gmail.com
mzhao4@dhu.edu.cn   hjzhang@hit.edu.cn

## A   More discussion about LSR and KLSR

Note that if $n \ll m$, using the *push-through identity* [Henderson and Searle, 1981], we reformulate $C = (X^\top X + \lambda I)^{-1} X^\top X$ as $C = X^\top (\lambda I + X X^\top)^{-1} X$ to reduce the computational cost from $O(n^3)$ to $O(mn^2)$. In $C = (K + \lambda I)^{-1} K$, when $n$ is large (e.g.$> 5000$), we perform randomized SVD [Halko *et al.*, 2011] on $K$: $K \approx V_r \Sigma_r V_r^\top$. Then $C \approx V_r \Sigma_r^{1/2} (\lambda I + \Sigma)^{-1} \Sigma_r^{1/2} V_r^\top$, where $r = 20k$ works well in practical applications. The time complexity of computing $C$ is $O(r\tau n + rn^2)$. The computation of the smallest $k + 1$ eigenvalues of $L$ is equivalent to compute the largest $k + 1$ eigenvalues and eigenvectors of $D^{-1/2} A D^{-1/2}$, which is sparse. The time complexity is $O(k\tau n)$. We have the following result.

**Proposition A.1.** *Let $\hat{c}$ be the optimal solution of* $\text{minimize}_c \frac{1}{2}\|\phi(y) - \phi(X)c\|^2 + \frac{\lambda}{2}\|c\|^2$, *where $\phi$ is induced by Gaussian kernel and $y$ is arbitrary. Then $\|\hat{c}_i - \hat{c}_j\| \leq \sqrt{2 - 2\exp\left(-\|x_i - x_j\|^2/(2\varsigma^2)\right)}$.*

It shows that when two data points in $X$, e.g. $x_i$ and $x_j$, are close to each other, the corresponding two elements in $\hat{c}$, e.g. $\hat{c}_i$ and $\hat{c}_j$, have small difference. Hence KLSR with Gaussian kernel utilizes local information to enhance $C$.

In LSR and KLSR, let $\lambda \in \Lambda$, $\tau \in \mathcal{T}$, and $\Theta = \Lambda \times \mathcal{T}$. The algorithm of AutoSC-GD with only LSR and KLSR is shown in Algorithm 1. The total time complexity is

$$O\left(|\Lambda|(mn^2 + r\bar{\tau}n + rn^2) + 2|\Lambda||\mathcal{T}|k\bar{\tau}n\right),$$

where $\bar{\tau}$ denotes the mean value in $\mathcal{T}$. The time complexity is at most $O(|\Lambda|\left(mn^2 + |\mathcal{T}|kmn\right))$ when $\tau \leq r \leq m \leq n$. It is worth noting that Algorithm 1 can be easily implemented parallelly, which will reduce the time complexity to $O(\max(m, r)n^2 + kmn)$. On the contrary, SSC, LRR, and their variants require iterative optimization and hence their time complexity is about $O(tmn^2)$, where $t$ denotes the iteration number and is often larger than 100.

## B   The algorithm of AutoSC+NSE

See Algorithm 2.

## C   More theoretical results

---

*Corresponding author

**Algorithm 1** AutoSC-GD with Only LSR and KLSR

**Input:** $\boldsymbol{X}, k, \mathcal{F}, \Lambda, \mathcal{T}$
 1: Normalize the columns of $\boldsymbol{X}$ to have unit $\ell_2$ norm.
 2: **for** $f_u$ in $\mathcal{F}$ **do**
 3:    **for** $\lambda_i$ in $\Lambda$ **do**
 4:       Construct $\boldsymbol{C}$ by (13) or (15).
 5:       **for** $\tau_j$ in $\mathcal{T}$ **do**
 6:          $\boldsymbol{C} \leftarrow |\boldsymbol{C} \odot (\boldsymbol{1} - \boldsymbol{I})|$.
 7:          Truncate $\boldsymbol{C}$ with parameter $\tau_j$.
 8:          For $j = 1, \ldots, n$, let $\boldsymbol{c}_j \leftarrow \boldsymbol{c}_j/|\boldsymbol{c}_j|_1$.
 9:          $\boldsymbol{A} = (\boldsymbol{C} + \boldsymbol{C}^\top)/2$.
10:          $\boldsymbol{L} = \boldsymbol{I} - \boldsymbol{D}^{-1/2}\boldsymbol{A}\boldsymbol{D}^{-1/2}$.
11:          Compute $\sigma_1, \ldots, \sigma_{k+1}$ and $\boldsymbol{v}_1, \ldots, \boldsymbol{v}_k$.
12:          $\Delta_{uij} = \mathrm{REG}(\boldsymbol{L})$, $\mathcal{V}_{uij} = [\boldsymbol{v}_1, \ldots, \boldsymbol{v}_k]$.
13:       **end for**
14:    **end for**
15: **end for**
16: $\boldsymbol{Z} = \mathcal{V}_{\bar{u}\bar{i}\bar{j}}^\top$, where $\{\bar{u}, \bar{i}, \bar{j}\} = \mathrm{argmax}_{u,i,j}\Delta_{uij}$.
17: Normalize the columns of $\boldsymbol{Z}$ to have unit $\ell_2$ norm.
18: Perform k-means on $\boldsymbol{Z}$.
**Output:** $k$ clusters: $C_1, \ldots, C_k$.

---

**Algorithm 2** AutoSC+NSE

**Input:** $\boldsymbol{X}, k, \mathcal{F}, \Theta, \hat{n}$.
 1: Select $\hat{n}$ landmarks from $\boldsymbol{X}$ by k-means to form $\hat{\boldsymbol{X}}$.
 2: Apply AutoSC-GD or AutoSC-BO to $\hat{\boldsymbol{X}}$ with $\mathcal{F}$ and $\Theta$.
 2: Get $\hat{\boldsymbol{Z}}$ from the best Laplacian matrix given by AutoSC-G or AutoSC-BO.
 3: Use mini-batch Adam to solve (17).
 4: Compute $\boldsymbol{Z}$ by (18).
 5: Perform k-means on $\boldsymbol{Z}$.
**Output:** $k$ clusters: $C_1, \ldots, C_k$.

## C.1   Theoretical guarantee for KLSR

**Definition C.1** (Polynomial Deterministic Model). The columns of $\boldsymbol{X}_0 \in \mathbb{R}^{m \times n}$ are drawn from a union of $k$ different polynomials $\{g_j : \mathbb{R}^r \to \mathbb{R}^m, \ r < m\}_{j=1}^k$ of order at most $p$ and are further corrupted by noise, say $\boldsymbol{X} = \boldsymbol{X}_0 + \boldsymbol{E}$. Denote the eigenvalue decomposition of the kernel matrix $\boldsymbol{K}$ of $\boldsymbol{X}$ as $\boldsymbol{K} = \boldsymbol{V}\boldsymbol{\Sigma}\boldsymbol{V}^\top$, where $\boldsymbol{\Sigma} = \mathrm{diag}(\sigma_1, \ldots, \sigma_n)$ and $\sigma_1 \geq \sigma_2 \geq \cdots \sigma_n$. Let $\gamma = \sigma_{d+1}/\sigma_d$. Denote $\boldsymbol{v}_i = (v_{i1}, \ldots, v_{in})$ the $i$-th row of $\boldsymbol{V}$ and let $\bar{\boldsymbol{v}}_i = (v_{i1}, \ldots, v_{id})$, where $d < n$. Suppose the following conditions hold: 1) for every $i \in [n]$, the $\bar{\tau}$-th largest element of $\{|\bar{\boldsymbol{v}}_i^\top \bar{\boldsymbol{v}}_j| : j \in C_{\pi(i)}\}$ is greater than $\alpha$; 2) $\max_{i \in [n]} \max_{j \in [n]\setminus C_{\pi(i)}} |\bar{\boldsymbol{v}}_i^\top \bar{\boldsymbol{v}}_j| \leq \beta$; 3) $\max_{i,j,l} |v_{il}v_{jl}| \leq \mu$.

Here we consider polynomials because they are easy to analyze and can well approximate smooth functions provided that $p$ is sufficiently large. Clustering the columns of $\boldsymbol{X}$ given by Definition C.1 according to the polynomials is actually a manifold clustering problem beyond the setting of subspace clustering. Similar to the subspace detection property, we define

**Definition C.2** (Manifold Detection Property). A symmetric affinity matrix $\boldsymbol{A}$ obtained from $\boldsymbol{X}$ has manifold detection property if for all $i$, the nonzero elements of $\boldsymbol{a}_i$ correspond to the columns of $\boldsymbol{X}$ lying on the same manifold as $\boldsymbol{x}_i$.

The following theorem verifies the effectiveness of (15) followed by the truncation operation in manifold detection.

**Theorem C.3.** *Suppose $\boldsymbol{X}$ and $\boldsymbol{K}$ are given by Definition C.1 and $\boldsymbol{C}$ is given by* (15)*, where the kernel function is a polynomial kernel of order $q$,* $\mathrm{rank}(\boldsymbol{K}_0) = d$ *($\boldsymbol{K}_0$ is from $\boldsymbol{X}_0$), and*

$$\frac{\left(\rho - \sqrt{\rho^2 - 4(2\mu d - \Delta)(2\mu n - 2\mu d - \Delta)}\right)\sigma_d^2}{4\mu d - 2\Delta} < \lambda < \frac{\left(\rho + \sqrt{\rho^2 - 4(2\mu d - \Delta)(2\mu n - 2\mu d - \Delta)}\right)\sigma_d^2}{4\mu d - 2\Delta} \tag{1}$$

*where $\rho = 2\mu n \gamma^2 - \Delta(1 + \gamma^2)$. Then $d \le k\binom{r+pq}{pq}$ and the $\boldsymbol{C}$ truncated by $\tau \le \bar{\tau}$ has the manifold detection property.*

In the theorem, $\sigma_d$ can be much larger than $\sigma_{d+1}$ provided that the noise is small enough. Then we get a wide range for $\lambda$. Compared to Theorem 3.7, Theorem C.3 allows a much larger $d$, which means the kernel method is able to handle more difficult clustering problems than the linear method.

## C.2 Theoretical analysis for NSE

The following proposition shows that a small number of hidden nodes in NSE are sufficient to make the clustering succeed.

**Proposition C.4.** *Suppose the columns (with unit $\ell_2$ norm) of $\boldsymbol{X}$ are drawn from a union of $k$ independent subspaces of dimension $r$: $\sum_{j=1}^{k} \dim(\mathcal{S}_j) = \dim(\mathcal{S}_1 \cup \cdots \cup \mathcal{S}_k) = kr$. For $j = 1, \ldots, k$, let $\boldsymbol{U}^j$ be the bases of $\mathcal{S}_j$ and $\boldsymbol{x}_i = \boldsymbol{U}^j \boldsymbol{v}_i$, if $\boldsymbol{x}_i \in \mathcal{S}_j$. Suppose $\max\{\|\boldsymbol{U}_{:l}^{i}{}^\top \boldsymbol{U}^j\| : 1 \le l \le r, 1 \le i \ne j \le k\} \le \mu$. Suppose that for all $i = 1, \ldots n$, $\max\{v_{1i}, \ldots, v_{ri}\} > \mu$. Then there exist $\boldsymbol{W}_1 \in \mathbb{R}^{d \times m}$, $\boldsymbol{W}_2 \in \mathbb{R}^{k \times d}$, $\boldsymbol{b}_1 \in \mathbb{R}^d$, and $\boldsymbol{b}_2 \in \mathbb{R}^k$ such that performing k-means on $\boldsymbol{Z}$ given by* (18) *identifies the clusters correctly, where $d = kr$.*

# D   More about the experiments

## D.1   Dataset description

The description for the benchmark image datasets considered in this paper are as follows.

- **Extended Yale B Face** [Kuang-Chih *et al.*, 2005] (Yale B for short): face images (192×168) of 38 subjects. Each subject has about 64 images under various illumination conditions. We resize the images into $32 \times 32$.

- **ORL Face** [Samaria and Harter, 1994]: face images (112×92) of 40 subjects. Each subject has 10 images with different poses and facial expressions. We resize the images into 32×32.

- **COIL20** [Nene *et al.*, 1996]: images ($32 \times 32$) of 20 objects. Each object has 72 images of different poses.

- **AR Face** [Martínez and Kak, 2001]: face images (165×120) of 50 males and 50 females. Each subject has 26 images with different facial expressions, illumination conditions, and occlusions. We resize the images into $42 \times 30$.

- **MNIST** [LeCun *et al.*, 1998]: 70,000 grey images ($28 \times 28$) of handwritten digits $0 - 9$.

- **MNIST-1k(10k)**: a subset of MNIST containing 1000(10000) samples, 100(1000) randomly selected samples per class.

- **Fashion-MNIST** [Xiao *et al.*, 2017]: 70,000 gray images ($28 \times 28$) of 10 types of fashion product.

- **Fashion-MNIST-1k(10k)**: a subset of Fashion-MNIST containing 1000(10000) samples, 100(1000) randomly selected samples per class.

- **MNIST-feature**: following the same procedures of [Chen *et al.*, 2020], we compute a feature vector of dimension 3,472 using the scattering convolution network [Bruna and Mallat, 2013] and then reduce the dimension to 500 using PCA.

- **Fashion-MNIST-feature**: similar to MNIST-feature.

- **GTSRB** [Stallkamp *et al.*, 2012]: consisting of 12,390 images of street signs in 14 categories. Following [Chen *et al.*, 2020], we extract a 1568-dimensional HOG feature, and reduce the dimension to 500 by PCA.

All experiments are conducted in MATLAB on a MacBook Pro with 2.3 GHz Intel i5 Core and 8GB RAM.

## D.2 Hyperparameter settings for the small datasets

We select $\lambda$ from $\{0.01, 0.02, 0.05, 0.1, 0.2, \ldots, 0.5\}$ for SSC, LRR, and KSSC. The $\lambda$ in BDR is chosen from $\{5, 10, 20, \ldots, 80\}$. The $\gamma$ in BDR-B and BDR-Z is chosen from $\{0.01, 0.1, 1\}$. The parameter $s$ in SSC-OMP is chosen from $\{3, 4, \ldots, 15\}$. We report the results of these methods with their best hyperparameters. In AutoSC, we set $\Lambda = \{0.01, 0.1, 1\}$ and $\mathcal{T} = \{5, 6, \ldots, 15\}$. In AutoSC-BO, we consider two models: 1) Gaussian kernel similarity; 2) KLSR with polynomial kernel; 3) KLSR with Gaussian kernel, in which the hyperparameters of kernels are optimized adaptively. Then we needn't to consider LSR explicitly because it is a special case of KLSR with polynomial kernel. See Appendix D.5.

## D.3 Clustering results in terms of NMI

In addition to the clustering accuracy reported in Table 4, here we also compare the normalized mutual information (NMI) in Table 1. We see that the comparative performance of all methods are similar to the results in Table 4 and our methods AutoSC-GD and AutoSC-BO outperformed other methods in almost all cases.

Table 1: Normalized Mutual Information on the six small datasets

|  | SSC | LRR | EDSC | KSSC | SSC-OMP | BDR-Z | BDR-B | AutoSC-GD | AutoSC-BO |
|---|---|---|---|---|---|---|---|---|---|
| Yale B | 0.817 | 0.703 | 0.835 | 0.730 | 0.841 | 0.666 | 0.743 | **0.919** | **0.928** |
| ORL | 0.849 | 0.872 | 0.856 | 0.872 | 0.815 | 0.875 | 0.865 | **0.907** | **0.903** |
| COIL20 | 0.954 | 0.706 | 0.843 | **0.983** | 0.671 | 0.843 | 0.873 | 0.897 | **0.963** |
| AR | 0.818 | 0.872 | 0.825 | 0.809 | 0.691 | 0.865 | 0.861 | **0.887** | **0.904** |
| MNIST-1k | 0.612 | 0.538 | 0.631 | 0.626 | 0.546 | 0.634 | 0.580 | **0.667** | 0.652 |
| Fashion-MNIST-1k | 0.616 | 0.601 | 0.621 | 0.621 | 0.559 | 0.614 | 0.605 | **0.633** | **0.629** |

## D.4 The stability of AutoSC

Though we have used a relatively compact search space in AutoSC to reduce the highly unnecessary computational cost, the search space can be arbitrarily large. Figure 1 shows the clustering accuracy and the corresponding relative-eigen-gap. We can see that the region with highest relative-eigen-gap is in accordance with the region with highest clustering accuracy.

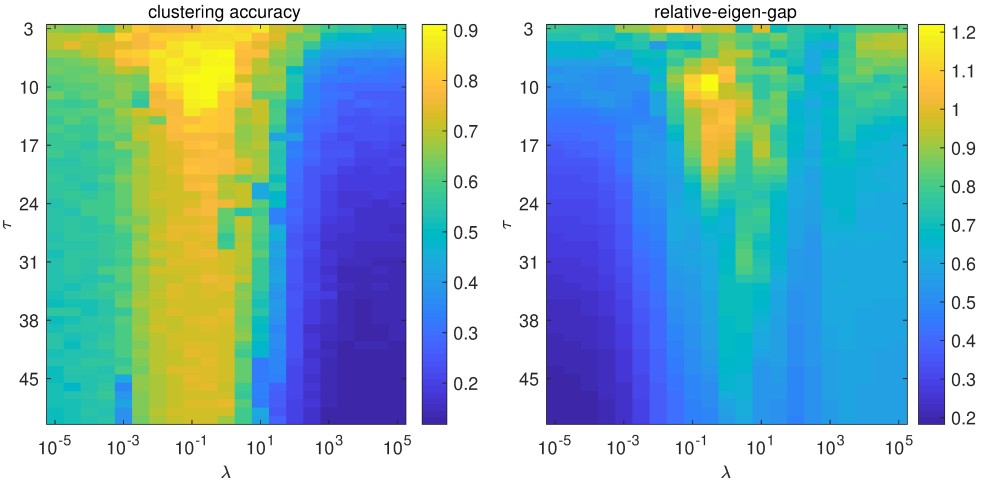

Figure 1: Visualization of the clustering accuracy and the corresponding relative-eigen-gap when a large search space is used.

## D.5 More about AutoSC-BO in the experiments

For SSC, we consider the following problem

$$\underset{C}{\text{minimize}} \ \tfrac{1}{2}\text{Tr}\left(\boldsymbol{K} - 2\boldsymbol{K}\boldsymbol{C} + \boldsymbol{C}^\top \boldsymbol{K}\boldsymbol{C}\right) + \lambda\|\boldsymbol{C}\|_1, \tag{2}$$

where $\boldsymbol{K}$ is an $n \times n$ kernel matrix with $[\boldsymbol{K}]_{ij} = k(\boldsymbol{x}_i, \boldsymbol{x}_j)$. Note that when we use a linear kernel function, (2) reduces to the vanilla SSC. We solve the optimization via alternating direction method of multipliers (ADMM) [Boyd *et al.*, 2011], where the Lagrange parameter is 0.1 and the maximum number of iterations is 500. In this study, we consider polynomial kernel and Gaussian kernel, and optimize all hyperparameters including the order of the polynomial kernel. Particularly, for Gaussian kernel, we set $\varsigma = \frac{\xi}{n^2}\sum_{ij}\|\boldsymbol{x}_i - \boldsymbol{x}_j\|$ and optimize $\xi$. The search space for the hyperparameters are as follows: $10^{-3} \leq \lambda \leq 1, 5 \leq \tau \leq 50, 0 \leq b \leq 10^3, 1 \leq q \leq 5, 0.5 \leq \xi \leq 5$.

In addition to Figure 2 of the main paper, here we report the best hyperparameters of the four models found by AutoSC-BO in Table 2. It can be found that the accuracy of KLSR with a linear kernel is higher than other models, which is consistent with its highest reg.

Table 2: The best hyperparameters and the corresponding clustering accuracy given by $\text{AutoSC}_{BO}$ on the first 10 subjects of YaleB Face dataset.

| method | hyperparameters | reg | accuracy |
|---|---|---|---|
| KLSR (Polynomial) | $\lambda = 0.207, b = 19.09, q = 1, \tau = 5$ | 2.379 | 0.966 |
| KLSR (Gaussian) | $\lambda = 0.013, \xi = 4.92, \tau = 5$ | 2.217 | 0.963 |
| KSSC (Polynomial) | $\lambda = 0.519, b = 44.57, q = 2, \tau = 5$ | 1.388 | 0.859 |
| KSSC (Gaussian) | $\lambda = 0.0011, \xi = 4.97, \tau = 6$ | 0.892 | 0.584 |

## D.6 Hyperparameter settings of large-scale clustering

On MNIST-10k, MNIST, Fashion-MNIST-10k, and Fashion-MNIST, the parameter settings of [Chen and Cai, 2011], SSSC [Peng *et al.*, 2013], SSC-OMP [You *et al.*, 2016], and S⁵C [Matsushima and Brbic, 2019], and S³COMP-C [Chen *et al.*, 2020], and AutoSC+NSE are shown in Table 3. These hyper parameters have been determined via grid search and the best (as possible) values are used.

Table 3: Hyper-parameter settings of the compared methods on MNIST-10k, MNIST, Fashion-MNIST-10k, and Fashion-MNIST. $s$ denotes the number of landmark data points. In the optimization (mini-batch Adam) of AutoSC+NSE, the epoch number, batch size, and step size are 200, 128, and $10^{-3}$ respectively.

| | |
|---|---|
| LSC-K | $s = 1000, r = 3$ |
| SSSC | $s = 1000, \lambda = 0.01$ |
| SSC-OMP | $K = 10$ (sparsity) |
| S⁵C | $s = 1000, \lambda = 0.1 \text{ or } 0.2$ |
| S³COMP-C | $T = 20, \lambda = 0.4, \delta = 0.9$ |
| AutoSC+NSE | $s = 1000, d = 200, \gamma = 10^{-5}$ |
| $\text{AutoSC}_{BO}$+NSE | $s = 1000, d = 200, \gamma = 10^{-5}$ |

## D.7 Influence of hyper-parameters in AutoSC+NSE

We investigate the effects of the type of activation function and the number ($d$) of nodes in the hidden layer of NSE. For convenience, we used a fixed random seed of MATLAB (rng(1)). Figure 2 shows the clustering accuracy on MNIST given by AutoSC+NSE with different activation function and different $d$. We see that ReLU outperformed tanh consistently. The reason is that the nonlinear

mapping $g$ from the data space to the eigenspace of the Laplacian matrix is nonsmooth and ReLU is more effective than tanh in approximating nonsmooth functions. In addition, when $d$ increases, the clustering accuracy of AutoSC+NSE with ReLU often becomes higher because a wider network often has a higher ability of function approximation.

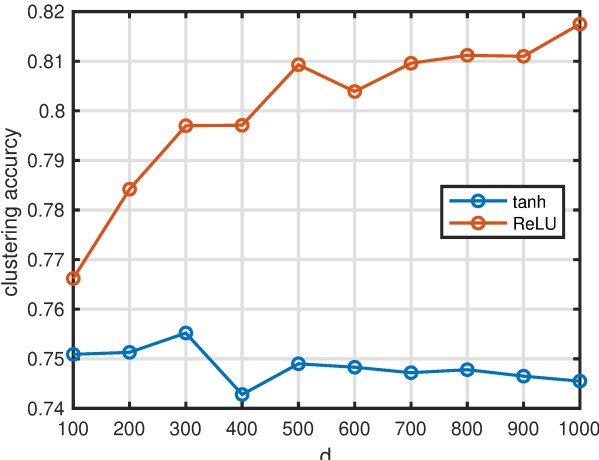

Figure 2: ReLU v.s. tanh (hyperbolic tangent) in the hidden layer of AutoSC+NSE on MNIST. When using ReLU, we set $\gamma = 10^{-5}$ and $\alpha = 10^{-3}$ (the step size in Adam). When using tanh, we set $\gamma = 10^{-3}$ and $\alpha = 10^{-2}$, which perform best in this case. Notice that the clustering accuracy when using ReLU is higher than 0.78 in almost all cases, which is higher than the value (say 0.755) we reported in the main paper. The reason is that in the main paper, we reported the mean value of 10 repeated trials but here we report the value of a single trial.

Figure 3 shows the clustering accuracy on MNIST given by AutoSC+NSE with different $\gamma$ and $\alpha$. When $\alpha$ is too small (say $10^{-4}$, the clustering accuracy is low, because the training error is quite large in 200 epochs. In fact, by increasing the training epochs, the clustering accuracy can be improved, which however will increase the time cost. When $\alpha$ is relatively large, the clustering accuracy is often higher than 0.755. On the other hand, AutoSC+NSE is not sensitive to $\gamma$ provided that it is not too large.

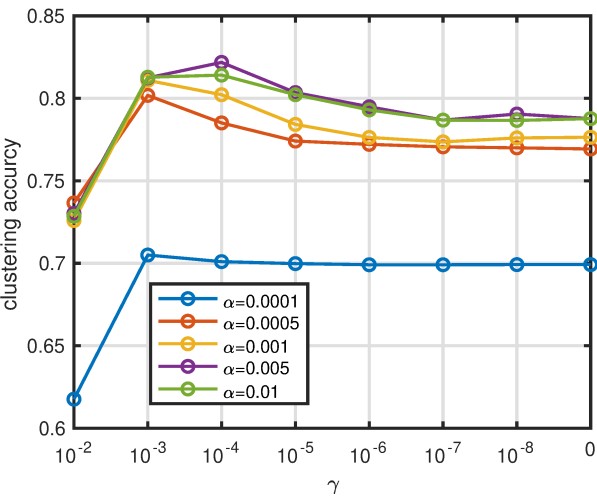

Figure 3: Influence of $\gamma$ and $\alpha$ in AutoSC+NSE on MNIST. We set $d = 200$ and use ReLU.

Figure 4 shows the mean value and standard deviation (10 repeated trials) of the clustering accuracy on MNIST given by AutoSC+NSE with different number (denoted by $s$) of landmark points. It can be found that when the $s$ increases, the clustering accuracy increases and its standard deviation becomes smaller. When $s$ is large enough, the improvement is not significant.

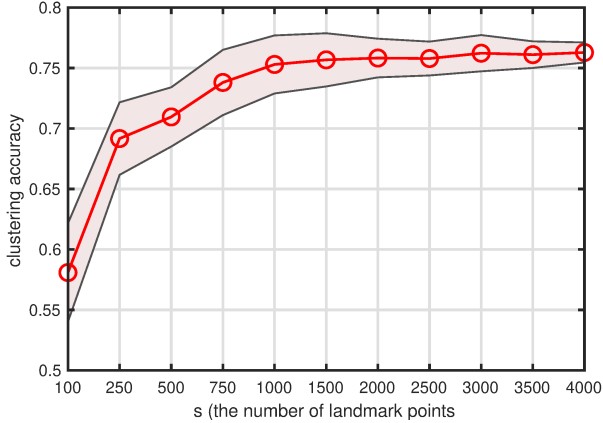

Figure 4: Influence of the number of landmark points in AutoSC+NSE on MNIST. We set $d = 200$, $\gamma = 10^{-5}$, and $\alpha = 10^{-3}$. The shadow denotes the standard deviation of 10 trials.

# E    Proof for the theoretical results

## E.1    Proof for Claim 3.2

*Proof.* The stochastic transition matrix of $G$ is defined as

$$\boldsymbol{P} = \boldsymbol{D}^{-1}\boldsymbol{A}. \tag{3}$$

In [Meila, 2001], it was showed that

$$\mathrm{MNCut}(\mathcal{C}) \geq k - \sum_{i=1}^{k} \varrho_i(\boldsymbol{P}), \tag{4}$$

where $\varrho_i(\boldsymbol{P})$ denotes the $i$-th largest eigenvalue of $\boldsymbol{P}$ and $1 = \varrho_1(\boldsymbol{P}) \geq \varrho_2(\boldsymbol{P}) \geq \cdots \varrho_k(\boldsymbol{P})$. According to Lemma 3 of [Meila, 2001], we have

$$\sigma_i(\boldsymbol{L}) = 1 - \varrho_i(\boldsymbol{P}), \quad \forall i = 1, \ldots, n. \tag{5}$$

Substituting (5) into (4), we have

$$\mathrm{MNCut}(\mathcal{C}) \geq \sum_{i=1}^{k} \sigma_i(\boldsymbol{L}). \tag{6}$$

$\square$

*Remark* E.1. $\mathcal{C}$ can be any partition of the nodes of $G$. Let $\mathcal{C}^*$ be the optimal partition. Then $\mathrm{MNCut}(\mathcal{C}^*) = \sum_{i=1}^{k} \sigma_i(\boldsymbol{L})$. If $\sum_{i=1}^{k} \sigma_i(\boldsymbol{L}) = 0$, there are no connections (edges) among $C_1^*, \ldots, C_k^*$.

## E.2    Proof for Claim 3.3

*Proof.* For $i = 1, \ldots, k$, we aim to partition $C_i$ into two subsets, denoted by $C_i^1$ and $C_i^2$. Then we define

$$\mathrm{MNCut}(C_i) = \frac{Cut(C_i^1, C_i^2)}{Vol(C_i^1)} + \frac{Cut(C_i^2, C_i^1)}{Vol(C_i^2)}. \tag{7}$$

It follows that

$$\mathrm{MNCut}(C_i) \geq \sum_{j=1}^{2} \sigma_j(\boldsymbol{L}_{C_i}) \geq \sigma_2(\boldsymbol{L}_{C_i}) = ac(C_i), \tag{8}$$

where $\boldsymbol{L}_{C_i}$ denotes the Laplacian matrix of $C_i$ an $i = 1, \ldots, k$. Since $\sigma_{k+1}(\boldsymbol{L}) = \min\{ac(C_1), \ldots, ac(C_k)\}$, we have

$$\min_{1 \leq i \leq k} \mathrm{MNCut}(C_i) \geq \sigma_{k+1}(\boldsymbol{L}). \tag{9}$$

Therefore, $\sigma_{k+1}(\boldsymbol{L})$ measures the least connectivity of $C_1, \ldots, C_k$. This finished the proof. $\square$

*Remark* E.2. When $\sigma_{k+1}(\boldsymbol{L})$ is large, the connectivity in each of $C_1, \ldots, C_k$ is strong. Otherwise, the connectivity in each of $C_1, \ldots, C_k$ is weak. When $\sigma_{k+1}(\boldsymbol{L}) = 0$, at least one of $C_1, \ldots, C_k$ contains at least two components, which means the nodes of $G$ can be partitioned into $k+1$ or more clusters.

### E.3 Proof for Theorem 3.4

*Proof.* According to Theorem 1 of [Meila *et al.*, 2005], we have

$$\text{dist}(\mathcal{C}, \mathcal{C}') < \frac{3\delta}{\sigma_{k+1}(\boldsymbol{L}) - \sigma_k(\boldsymbol{L})}. \tag{10}$$

Since $\text{reg}(\boldsymbol{L}) = \dfrac{\sigma_{k+1}(\boldsymbol{L}) - \frac{1}{k}\sum_{i=1}^{k}\sigma_i(\boldsymbol{L})}{\frac{1}{k}\sum_{i=1}^{k}\sigma_i(\boldsymbol{L}) + \epsilon}$, we have

$$\sigma_{k+1}(\boldsymbol{L}) - \sigma_k(\boldsymbol{L}) = \text{reg}(\boldsymbol{L})(\bar{\sigma} + \epsilon) + \bar{\sigma} - \sigma_k(\boldsymbol{L}), \tag{11}$$

where $\bar{\sigma} = \frac{1}{k}\sum_{i=1}^{k}\sigma_i(\boldsymbol{L}) \geq \epsilon$. Invoking (11) into (10), we arrive at

$$
\begin{aligned}
\text{dist}(\mathcal{C}, \mathcal{C}') &< \frac{3\delta}{\text{reg}(\boldsymbol{L})(\bar{\sigma} + \epsilon) + \bar{\sigma} - \sigma_k(\boldsymbol{L})} \\
&\leq \frac{3\delta}{2\epsilon\,\text{reg}(\boldsymbol{L}) + \bar{\sigma} - k\bar{\sigma}} \\
&\leq \frac{3\delta}{2\epsilon\,\text{reg}(\boldsymbol{L}) + (1-k)\eta\epsilon} \\
&\leq \frac{1.5\delta\epsilon^{-1}}{\text{reg}(\boldsymbol{L}) + (1-k)\eta/2}.
\end{aligned}
$$

This finished the proof. $\qquad\square$

### E.4 Proof for Proposition A.1

*Proof.* Since $\hat{\boldsymbol{c}}$ is the optimal solution, we have

$$\phi(\boldsymbol{x}_i)^\top (\phi(\boldsymbol{y}) - \phi(\boldsymbol{X})\hat{\boldsymbol{c}}) + \lambda\hat{c}_i = 0,$$

$$\phi(\boldsymbol{x}_j)^\top (\phi(\boldsymbol{y}) - \phi(\boldsymbol{X})\hat{\boldsymbol{c}}) + \lambda\hat{c}_j = 0.$$

It follows that

$$
\begin{aligned}
\|\hat{c}_i - \hat{c}_j\| &= \| (\phi(\boldsymbol{x}_i) - \phi(\boldsymbol{x}_j))^\top (\phi(\boldsymbol{y}) - \phi(\boldsymbol{X})\hat{\boldsymbol{c}}) \| \\
&\leq \|\phi(\boldsymbol{x}_i) - \phi(\boldsymbol{x}_j)\| \|\phi(\boldsymbol{y}) - \phi(\boldsymbol{X})\hat{\boldsymbol{c}}\| \\
&= \sqrt{k(\boldsymbol{x}_i, \boldsymbol{x}_i) - 2k(\boldsymbol{x}_i, \boldsymbol{x}_j) + k(\boldsymbol{x}_j, \boldsymbol{x}_j)} \\
&\quad \times \|\phi(\boldsymbol{y}) - \phi(\boldsymbol{X})\hat{\boldsymbol{c}}\| \\
&= \sqrt{2 - 2k(\boldsymbol{x}_i, \boldsymbol{x}_j)} \|\phi(\boldsymbol{y}) - \phi(\boldsymbol{X})\hat{\boldsymbol{c}}\| \\
&\leq \sqrt{2 - 2k(\boldsymbol{x}_i, \boldsymbol{x}_j)} \|\phi(\boldsymbol{y})\| \\
&= \sqrt{2 - 2\exp\left(-\frac{\|\boldsymbol{x}_i - \boldsymbol{x}_j\|^2}{2\varsigma^2}\right)}.
\end{aligned}
\tag{12}
$$

In the second and last equalities, we used the fact that $\|\phi(\boldsymbol{y})\| = \|\phi(\boldsymbol{x})\| = 1$. In the second inequality, we used the fact that $\frac{1}{2}\|\phi(\boldsymbol{y}) - \phi(\boldsymbol{X})\hat{\boldsymbol{c}}\|^2 + \frac{\lambda}{2}\|\hat{\boldsymbol{c}}\|^2 \leq \frac{1}{2}\|\phi(\boldsymbol{y}) - \phi(\boldsymbol{X})\boldsymbol{0}\|^2 + \frac{\lambda}{2}\|\boldsymbol{0}\|^2 = \frac{1}{2}\|\phi(\boldsymbol{y})\|^2$ because $\hat{\boldsymbol{c}}$ is the optimal solution. $\qquad\square$

## E.5 Proof for Theorem 3.7

*Proof.* Invoking the SVD of $\boldsymbol{X}$ into the closed-form solution of LSR, we get

$$\boldsymbol{C} = \boldsymbol{V}\mathrm{diag}\left(\frac{\sigma_1^2}{\sigma_1^2 + \lambda}, \ldots, \frac{\sigma_n^2}{\sigma_n^2 + \lambda}\right)\boldsymbol{V}^\top. \tag{13}$$

It means

$$
\begin{aligned}
c_{it} &= \sum_{l=1}^{n} \frac{v_{il}v_{jl}\sigma_l^2}{\sigma_l^2 + \lambda} \\
&= \bar{\boldsymbol{v}}_i^\top \bar{\boldsymbol{v}}_t - \sum_{l=1}^{d} \frac{v_{il}v_{tl}\lambda}{\sigma_l^2 + \lambda} + \sum_{l=d+1}^{n} \frac{v_{il}v_{tl}\sigma_l^2}{\sigma_l^2 + \lambda}.
\end{aligned}
\tag{14}
$$

Suppose $j \in C_{\pi(i)}$ and $k \in [n] \setminus C_{\pi(i)}$. We have

$$
\begin{aligned}
&|c_{ij}| - |c_{ik}| \\
&= \left| \bar{\boldsymbol{v}}_i^\top \bar{\boldsymbol{v}}_j - \sum_{l=1}^{d} \frac{v_{il}v_{jl}\lambda}{\sigma_l^2 + \lambda} + \sum_{l=d+1}^{n} \frac{v_{il}v_{jl}\sigma_l^2}{\sigma_l^2 + \lambda} \right| \\
&\quad - \left| \bar{\boldsymbol{v}}_i^\top \bar{\boldsymbol{v}}_k - \sum_{l=1}^{d} \frac{v_{il}v_{kl}\lambda}{\sigma_l^2 + \lambda} + \sum_{l=d+1}^{n} \frac{v_{il}v_{kl}\sigma_l^2}{\sigma_l^2 + \lambda} \right| \\
&\geq \left| \bar{\boldsymbol{v}}_i^\top \bar{\boldsymbol{v}}_j \right| - \left| \bar{\boldsymbol{v}}_i^\top \bar{\boldsymbol{v}}_k \right| - \left| \sum_{l=1}^{d} \frac{v_{il}v_{jl}\lambda}{\sigma_l^2 + \lambda} \right| - \left| \sum_{l=1}^{d} \frac{v_{il}v_{kl}\lambda}{\sigma_l^2 + \lambda} \right| \\
&\quad - \left| \sum_{l=d+1}^{n} \frac{v_{il}v_{jl}\sigma_l^2}{\sigma_l^2 + \lambda} \right| - \left| \sum_{l=d+1}^{n} \frac{v_{il}v_{kl}\sigma_l^2}{\sigma_l^2 + \lambda} \right| \\
&\geq \left| \bar{\boldsymbol{v}}_i^\top \bar{\boldsymbol{v}}_j \right| - \left| \bar{\boldsymbol{v}}_i^\top \bar{\boldsymbol{v}}_k \right| - 2\mu \sum_{l=1}^{d} \frac{\lambda}{\sigma_l^2 + \lambda} - 2\mu \sum_{l=d+1}^{n} \frac{\sigma_l^2}{\sigma_l^2 + \lambda} \\
&\geq \left| \bar{\boldsymbol{v}}_i^\top \bar{\boldsymbol{v}}_j \right| - \beta - \frac{2\mu d\lambda}{\sigma_d^2 + \lambda} - \frac{2\mu a\sigma_{d+1}^2}{\sigma_{d+1}^2 + \lambda},
\end{aligned}
\tag{15}
$$

where $a = \min(m, n) - d = m - d$.

To ensure that there exist at least $\bar{\tau}$ elements of $\{|c_{ij}| : j \in C_{\pi(i)}\}$ greater than $|c_{ik}|$ for all $k \in [n]\setminus C_{\pi(i)}$, we need

$$\left| \bar{\boldsymbol{v}}_i^\top \bar{\boldsymbol{v}}_j \right| - \beta - \frac{2\mu d\lambda}{\sigma_d^2 + \lambda} - \frac{2\mu a\sigma_{d+1}^2}{\sigma_{d+1}^2 + \lambda} > 0 \tag{16}$$

holds at least for $\bar{\tau}$ different $j$, where $j \in C_{\pi(i)}$. It is equivalent to ensure that

$$\alpha - \beta - \frac{2\mu d\lambda}{\sigma_d^2 + \lambda} - \frac{2\mu a\sigma_{d+1}^2}{\sigma_{d+1}^2 + \lambda} > 0. \tag{17}$$

We rewrite (17) as

$$u_1\lambda^2 + u_2\lambda + u_3 > 0, \tag{18}$$

where $u_1 = \alpha - \beta - 2\mu d$, $u_2 = (\alpha - \beta)(\sigma_d^2 + \sigma_{d+1}^2) - 2\mu(d+a)\sigma_{d+1}^2$, and $u_3 = (\alpha - \beta - 2\mu a)\sigma_d^2\sigma_{d+1}^2$.

The definition of $\mu$, $\alpha$, and $\beta$ imply $u_1 < 0$. Then we solve (18) and obtain

$$
\begin{cases}
\lambda > \frac{2\mu m\sigma_{d+1}^2 - (\alpha-\beta)(\sigma_d^2 + \sigma_{d+1}^2) - \sqrt{w}}{2(2\mu d - (\alpha-\beta))} \\
\lambda < \frac{2\mu m\sigma_{d+1}^2 - (\alpha-\beta)(\sigma_d^2 + \sigma_{d+1}^2) + \sqrt{w}}{2(2\mu d - (\alpha-\beta))}
\end{cases}
\tag{19}
$$

where $w = u_2^2 - 4u_1u_3$. To simplify the notations, we let $\Delta = \alpha - \beta$, $\sigma_{d+1} = \gamma\sigma_d$ and get

$$
\begin{cases}
\lambda > \frac{\left(2\mu m\gamma^2 - \Delta(1+\gamma^2) - \sqrt{(\Delta(1+\gamma^2) - 2\mu m\gamma^2)^2 - 4(\Delta - 2\mu d)(\Delta - 2\mu m + 2\mu d)}\right)\sigma_d^2}{4\mu d - 2\Delta} \\
\lambda < \frac{\left(2\mu m\gamma^2 - \Delta(1+\gamma^2) + \sqrt{(\Delta(1+\gamma^2) - 2\mu m\gamma^2)^2 - 4(\Delta - 2\mu d)(\Delta - 2\mu m + 2\mu d)}\right)\sigma_d^2}{4\mu d - 2\Delta}
\end{cases}
\tag{20}
$$

Further, let $\rho = 2\mu m \gamma^2 - \Delta(1 + \gamma^2)$, we arrive at

$$\begin{cases} \lambda > \frac{\left(\rho - \sqrt{\rho^2 - 4(\Delta - 2\mu d)(\Delta - 2\mu m + 2\mu d)}\right)\sigma_d^2}{4\mu d - 2\Delta} \\ \lambda < \frac{\left(\rho + \sqrt{\rho^2 - 4(\Delta - 2\mu d)(\Delta - 2\mu m + 2\mu d)}\right)\sigma_d^2}{4\mu d - 2\Delta} \end{cases} \tag{21}$$

That means, if (21) holds, for every $i$, the indices of the largest $\bar{\tau}$ absolute elements in the $i$-th column of $C$ are in $C_{\pi(i)}$. Therefore, the truncation operation with parameter $\tau \leq \bar{\tau}$ ensures the subspace detection property. This finished the proof.

$\square$

## E.6 Proof for Proposition 3.8

*Proof.* The condition of reg means

$$\frac{\sigma_{k+1}(\boldsymbol{L}) - \frac{1}{k}\sum_{i=1}^{k}\sigma_i(\boldsymbol{L})}{\frac{1}{k}\sum_{i=1}^{k}\sigma_i(\boldsymbol{L}) + \epsilon} = \frac{\sigma_{k+1}(\boldsymbol{L})}{\epsilon} > 0.$$

For convenience, denote $\vartheta = \frac{1}{k}\sum_{i=1}^{k}\sigma_i(\boldsymbol{L})$. We have

$$-\vartheta\epsilon = \vartheta\sigma_{k+1}.$$

It indicates $\vartheta = 0$ and $\sigma_{k+1} \neq 0$. Therefore the graph has exactly $k$ connected components. Since the subspace or manifold detection property hold for $\boldsymbol{A}$, each component of $G$ is composed of the columns of $\boldsymbol{X}$ in the same subspace or manifold. Thus, all the columns of $\boldsymbol{X}$ in the same subspace or manifold must be in the same component. Otherwise, the number of connected components is larger than $k$. $\square$

## E.7 Proof for Theorem C.3

The proof is nearly the same as that for Theorem 3.7, except that $d < n$ and $\text{rank}(\boldsymbol{K}_0) \leq k\binom{r+pq}{pq}$, where $\boldsymbol{K}_0 = \phi(\boldsymbol{X}_0)^\top \phi(\boldsymbol{X}_0)$. In this case, $\boldsymbol{K}$ can be well approximately by a low-rank matrix of rank at most $k\binom{r+pq}{pq}$ provided that the noise is small enough. More details about $\boldsymbol{K}_0$ can be found in [Fan *et al.*, 2020].

## E.8 Proof for Proposition C.4

*Proof.* We only need to provide an example of $\boldsymbol{W}_1 \in \mathbb{R}^{d \times m}$, $\boldsymbol{W}_2 \in \mathbb{R}^{k \times d}$, $\boldsymbol{b}_1 \in \mathbb{R}^d$, and $\boldsymbol{b}_2 \in \mathbb{R}^k$, where $d = kr$, such that the clusters can be recognized by k-means.

We organize the rows of $\boldsymbol{W}_1$ into $k$ groups: $\boldsymbol{W}_1^j \in \mathbb{R}^{r \times m}$, $j = 1, \ldots, k$. Let $\boldsymbol{W}_1^j = \boldsymbol{U}^{j\top}$, $j = 1, \ldots, k$. Let $\boldsymbol{W}_1 \boldsymbol{x}_i = \boldsymbol{\alpha}_i = (\boldsymbol{\alpha}_i^1, \ldots, \boldsymbol{\alpha}_i^r)$. When $\boldsymbol{x}_i \in \mathcal{S}_j$, we have

$$\boldsymbol{\alpha}_i^j = \boldsymbol{U}^{j\top} \boldsymbol{x}_i = \boldsymbol{U}^{j\top} \boldsymbol{U}^j \boldsymbol{v}_i = \boldsymbol{v}_i. \tag{22}$$

It follows from the assumption that

$$\max_p \alpha_{pi}^j > \mu. \tag{23}$$

Let $\boldsymbol{b}_1 = [\boldsymbol{b}_1^1; \ldots; \boldsymbol{b}_1^k] = -\mu\mathbf{1}$. Then $\boldsymbol{h}_i^j = \text{ReLU}(\boldsymbol{\alpha}_i^j + \boldsymbol{b}_1^j)$ has at least one positive element. On the other hand, since

$$\boldsymbol{\alpha}_i^l = \boldsymbol{U}^{l\top} \boldsymbol{x}_i = \boldsymbol{U}^{l\top} \boldsymbol{U}^j \boldsymbol{v}_i \quad l \neq j, \tag{24}$$

using the assumption of $\mu$, we have

$$|\alpha_{pi}^l| = |\boldsymbol{U}_{:p}^{l\top} \boldsymbol{U}^j \boldsymbol{v}_i| \leq \|\boldsymbol{U}_{:p}^{l\top} \boldsymbol{U}^j\| \|\boldsymbol{v}_i\| \leq \mu, \tag{25}$$

where we have used the fact $\|\boldsymbol{v}_i\| = 1$ because $\|\boldsymbol{x}_i\| = 1$. It follows that

$$\boldsymbol{h}_i^l = \text{ReLU}(\boldsymbol{\alpha}_i^l + \boldsymbol{b}_1^l) = \mathbf{0}, \quad l \neq j.$$

Now we formulate $\boldsymbol{W}_2$ as

$$\boldsymbol{W}_2 = \begin{bmatrix} \boldsymbol{q}_{11} & \boldsymbol{q}_{12} & \dots & \boldsymbol{q}_{1k} \\ \boldsymbol{q}_{21} & \boldsymbol{q}_{22} & \dots & \boldsymbol{q}_{2k} \\ \vdots & \vdots & \ddots & \vdots \\ \boldsymbol{q}_{k1} & \boldsymbol{q}_{k2} & \dots & \boldsymbol{q}_{kk} \end{bmatrix}, \tag{26}$$

where $\boldsymbol{q}_{lj} \in \mathbb{R}^{1 \times r}$, $l, j = 1, \dots, k$. We have

$$z_{ji} = \boldsymbol{q}_{j1} \boldsymbol{h}_i^1 + \boldsymbol{q}_{j2} \boldsymbol{h}_i^2 \cdots + \boldsymbol{q}_{jk} \boldsymbol{h}_i^k = \boldsymbol{q}_{jj} \boldsymbol{h}_i^j.$$

and

$$z_{li} = \boldsymbol{q}_{l1} \boldsymbol{h}_i^1 + \boldsymbol{q}_{l2} \boldsymbol{h}_i^2 \cdots + \boldsymbol{q}_{lk} \boldsymbol{h}_i^k = \boldsymbol{q}_{lj} \boldsymbol{h}_i^j.$$

Here we have let $\boldsymbol{b}_2 = \boldsymbol{0}$. Let $\boldsymbol{q}_{jj} \geq \boldsymbol{0}$ and $\boldsymbol{q}_{lj} = \boldsymbol{0}$, we have

$$z_{ji} > z_{li} = 0.$$

Therefore, if $\boldsymbol{x}_i \in \mathcal{S}_j$, we have $z_{ji} > 0$ and $z_{li} = 0 \ \forall 1 \leq j \neq l \leq k$. Now performing k-means on $\boldsymbol{Z} = [\boldsymbol{z}_1, \dots, \boldsymbol{z}_n]$ can identify the clusters trivially.

$\square$