# OpenReview forum: "A Simple Approach to Automated Spectral Clustering"
_NeurIPS.cc/2022/Conference — NeurIPS 2022 Accept_

### Official Review · Reviewer_37XY · 2022-07-06

**Rating:** 8
**Confidence:** 5
**Soundness:** 4 excellent
**Presentation:** 3 good
**Contribution:** 4 excellent

**Summary:**

The paper proposes a method of automated spectral clustering that is able to adaptively select hyper-parameters for spectral clustering. Specifically, the authors first present an optimization metric called relative eigen-gap and then use Bayesian optimization to maximize the metric to find suitable models and hyper-parameters for spectral clustering. To improve the efficiency as well as the accuracy, the authors propose to use least squares representation with thresholding to construct the adjacency matrix and prove the effectiveness. The experiments show that the proposed method is able to outperform a few baselines such as SSC and BDR.

**Questions:**

1. In equation (4), the definition of $f_{\theta}$ is not very clear or intuitive. The authors may provide an example for $f$ and $\theta$.

2. It was claimed that the $\varepsilon$ in equation (6) is a small constant such as 1e-6. What if it is too big?

3. In line 154, J(L) is different from the one defined by (Meila and Shortreed 2006), unless $\alpha\leq 0$. In line 159, J(L) with $\alpha=0$ is just a special case of (Meila and Shortreed 2006). I guess it is a typo and should be $\alpha\leq 0$, according to Table 3 in the appendix.

4. In line 175, the authors claimed that the DSC [Ji et al., 2017] method can be incorporated into the proposed AutoSC-BO. Do the authors mean that we can update the hyperparameter $\lambda$ of DSC using AutoSC-BO? How about the network structure and learning rate? I think it would be very time-consuming when inserting DSC into AutoSC-BO.

5. In Theorem 4.4, it seems that $\lambda$ does not exist if $2\mu d\leq \Delta$. The authors may provide more explanation for this.

6. In Tables 4 and 5, did the authors use the same NSE network structure for different datasets? If not, one may worry that NSE increases the difficulty of model and hyper-parameter selection in clustering.


**Strengths And Weaknesses:**

Strengths:
1. Model and hyper-parameter selection for unsupervised learning especially clustering is a challenging problem. Previous work on this problem is very limited. The paper provided a simple yet effective method, which is interesting and practical. The authors also proved the effectiveness of the relative-eigen-gap using Theorem 3.4.

2. The authors proved the effectiveness of least squares representation with thresholding in subspace (or manifold) detection, though least squares representation has been used in previous work of subspace clustering. To my knowledge, the theoretical results of Theorem 4.4 and Theorem C.3 look solid. They verified the roles of $\lambda$ and $\tau$.

3. The authors provided a novel method NSE for large-scale clustering.

4. The literature is sufficient and the related work has been clearly discussed.

5. The experiments were well designed and the results are plentiful. For instance, Figure 1 showed the impact of $\lambda$, $\tau$, and kernel in least squares representation. Table 2 compared different adjacency matrices and verified the connection between clustering accuracy and relative eigen-gap. Tables 4 and 5 demonstrated the effectiveness of the proposed methods in large-scale clustering.

In sum, the paper is well-written and has both theoretical guarantees and numerical verification. In my opinion, it solved a challenging problem in clustering.

Weaknesses:
1. Some notations (e.g. $\sigma_1,\sigma_2,\ldots$ in Table 1 and $L_s$ in Section 3.3) were not clearly explained.

2. It is not clear how efficient when AutoSC is applied to deep learning based clustering.

3. The conclusion section is a little bit short. Please provide more discussion.

---

> ### Author Response · Authors · 2022-07-29
> **Thank you for this recognition of our work. We have enhanced some details of the work.**
>
> $\textbf{On the weaknesses}$
>
> 1 Some notations (e.g.  in Table 1 and  $L_s$ in Section 3.3) were not clearly explained.
>
> $\textbf{Response}$:  We added more explanations. There is a typo. $L_s$ should be $L$.
>
> 2 It is not clear how efficient when AutoSC is applied to deep learning based clustering.
>
> $\textbf{Response}$:  We tried to run AutoSC on the DSC method proposed by [Ji et al. 2017]. It needs more than 20 hours. Therefore, it is very necessary to consider fast and effective AMC method (e.g. LSR and KLSR) in AutoSC.
>
> 3 The conclusion section is a little bit short. Please provide more discussion.
>
> $\textbf{Response}$: We included the limitation in the conclusion section.
>
> $\textbf{Response to the questions}$
>
> 1 In equation (4), the definition of $f_\theta$ is not very clear or intuitive. The authors may provide an example for $f$ and $\theta$.
>
> $\textbf{Response}$: $f_\theta$ denotes a function with parameter $\theta$. For example, $\mathbf{A}=f_{\theta}(\mathbf{X})=[\exp(-\Vert \mathbf{x}_i-\mathbf{x}_j\Vert^2/(2\varsigma^2))]-\mathbf{I}$, where $\theta=\varsigma$.
>
> 2 It was claimed that the $\varepsilon$ in equation (6) is a small constant such as 1e-6. What if it is too big?
>
> $\textbf{Response}$: We use $\varepsilon$ to ensure numerical stability. A too big $\varepsilon$ will introduce a big bias to reg, which reduces the effectiveness of reg in comparing different affinity matrices.
>
> 3 In line 154, J(L) is different from the one defined by (Meila and Shortreed 2006), unless $\alpha\leq 0$. In line 159, J(L) with $\alpha=0$ is just a special case of (Meila and Shortreed 2006). I guess it is a typo and should be $\alpha\leq 0$, according to Table 3 in the appendix.
>
> $\textbf{Response}$: Yes. It is a typo. Thanks for pointing out.
>
> 4 In line 175, the authors claimed that the DSC [Ji et al., 2017] method can be incorporated into the proposed AutoSC-BO. Do the authors mean that we can update the hyperparameter  of DSC using AutoSC-BO? How about the network structure and learning rate? I think it would be very time-consuming when inserting DSC into AutoSC-BO.
>
> $\textbf{Response}$: Yes. We can use AutoSC-BO to tune the network structures and hyperparameters of deep learning based clustering methods such as DSC. However, it is very time consuming.
>
> 5 In Theorem 4.4, it seems that  $\lambda$ does not exist if $2\mu d\leq \Delta$. The authors may provide more explanation for this.
>
> $\textbf{Response}$:  According to the definition of $\alpha$ and $\beta$, $2\mu d>\Delta$ always holds.
>
> 6 In Tables 4 and 5, did the authors use the same NSE network structure for different datasets? If not, one may worry that NSE increases the difficulty of model and hyper-parameter selection in clustering.
>
> $\textbf{Response}$: Yes. We have used the same network structure for the datasets. Actually, the method is not sensitive to the network structure because AutoSC can find a very effective and stable affinity matrix.

---

> > ### Comment · Reviewer_37XY · 2022-08-05
> > **Update after rebuttal**
> >
> > The authors have answered my questions in the rebuttal. After checking the rebuttal to other's comments,  I will keep my initial rating and vote for acceptance, due to the clear motivation and contributions.  The authors may also provide the Python codes for academic use.

---

### Official Review · Reviewer_uu35 · 2022-07-11

**Rating:** 7
**Confidence:** 5
**Soundness:** 4 excellent
**Presentation:** 3 good
**Contribution:** 3 good

**Summary:**

The paper aims at automated spectral clustering. It has the following contributions: 1) it provided a relative eigen gap with some theoretical analysis for model and parameter selection in spectral clustering; 2) it proposed to use least square regression and truncation to obtain affinity matrices and showed that the method has subspace detection property; 3) it provided a method called NSE to improve the scalability of AutoSC on large datasets. Besides the theoretical analysis, the paper used extensive numerical results to show the effectiveness of the proposed methods. In general, the paper is interesting and easy to follow.

**Questions:**

In addition to the weaknesses mentioned above, it would be helpful if the authors could address the following questions.

1. In the current paper, the authors only studied the deterministic cases (e.g. Definition 4.3). A few previous works such as [Wang and Xu, 2013] and [Matsushima and Brbic, 2019] have studied the non-deterministic cases of subspace clustering. Is there any difficulty to provide some theoretical results for the non-deterministic cases?
2. Can the authors explain more about the meaning of $\gamma$ and $\beta$ defined in Definition 4.3? Is there any assumption about the noise?
3. In Proposition 4.5, is reg(L)>0 necessary?
4. In Figure 1(i), what does the green circle mean?
5. In Table 2, what’s the meaning of reg_max? In addition, it is not clear how the authors use ‘Polynomial kernel’ to construct the affinity matrix. The polynomial kernel may lead to negative values if the degree is odd.
6. In Table 3, AutoSC-BO is much slower than AutoSC-GD. The authors mentioned that they optimized all hyperparameters. What hyperparameters are? Please provide some details.
7. In Line 290, if NSE is ablated, what’s the accuracy of AutoSC-GD on the full MNIST?


**Limitations:**

The authors did not mention the limitation.

**Strengths And Weaknesses:**

Strengths:

1. The paper is interesting and the problem here addressed is important for the machine learning community.
2. The overall approach is well described and seems novel.
3.  The proposed method works well on the evaluated datasets.

Weaknesses:
1. The notation in Section 3.3 can be improved.
2. Some important results (e.g. Theorem C.3) are in the supplementary material rather than in the main paper.
3. The authors only reported the clustering accuracy, it should be better to have more commonly used clustering metrics such as NMI and ARI.
4. The text in Figure 2(ii) is too tiny.

---

> ### Author Response · Authors · 2022-07-29
> **We have improved the notation and organization and added more explanations.**
>
> $\textbf{On the weakness}$
>
> 1 The notation in Section 3.3 can be improved.
>
> $\textbf{Response}$: We have improved the notations. For example, we corrected the typo $\mathbf{L}_\theta$.
>
> 2 Some important results (e.g. Theorem C.3) are in the supplementary material rather than in the main paper.
>
> $\textbf{Response}$: Because of the page length limitation, we have no space for these results (e.g. Theorem C.3). By the way, Theorem C.3 is similar to Theorem 3.7 (in the revised paper) and we want to save the space.
>
> 3 The authors only reported the clustering accuracy, it should be better to have more commonly used clustering metrics such as NMI and ARI.
>
> $\textbf{Response}$: Actually, we have reported the NMI results in Table 2 of the supplementary material. Since the datasets considered in this paper are class-balanced datasets, the relative performance of ACC and NMI are similar.
>
> 4 The text in Figure 2(ii) is too tiny.
>
> $\textbf{Response}$:  We have enlarged the text.
>
> $\textbf{Response to the questions}$
>
> 1 In the current paper, the authors only studied the deterministic cases (e.g. Definition 4.3). A few previous works such as [Wang and Xu, 2013] and [Matsushima and Brbic, 2019] have studied the non-deterministic cases of subspace clustering. Is there any difficulty to provide some theoretical results for the non-deterministic cases?
>
> $\textbf{Response}$: Thanks for pointing out. Currently, it is quite difficult to prove the subspace detection property for the non-deterministic cases. The difficulty mainly arises from the fact that the least square representation does not generate a sparse affinity matrix.
>
> 2 Can the authors explain more about the meaning of $\gamma$ and $\beta$ defined in Definition 4.3? Is there any assumption about the noise?
>
> $\textbf{Response}$: The definition of $\gamma$ indicates that it can be regarded as a measure of the noise-signal-ratio. A smaller $\gamma$ indicates that the noise is smaller, then the clustering problem is easier and we can get a larger range for $\lambda$ according to Theorem 3.7.
>
> 3 In Proposition 4.5, is reg(L)>0 necessary?
>
> $\textbf{Response}$: Yes. It is necessary. If reg(L) is zero, there can be more than k components in the graph.
>
> 4 In Figure 1(i), what does the green circle mean?
>
> $\textbf{Response}$: We use the green circle to highlight the maximum values of reg given by LSR and KLSR.
>
> 5 In Table 2, what’s the meaning of reg_max? In addition, it is not clear how the authors use ‘Polynomial kernel’ to construct the affinity matrix. The polynomial kernel may lead to negative values if the degree is odd.
>
> $\textbf{Response}$: reg_max means the maximum value of the relative-eigen-gap found by AutoSC-BO for each AMC method.
>
> 6 In Table 3, AutoSC-BO is much slower than AutoSC-GD. The authors mentioned that they optimized all hyperparameters. What hyperparameters are? Please provide some details.
>
> $\textbf{Response}$: The hyperparameters includes $\lambda$, $\tau$, and kernel parameters (e.g. $b,q,\varsigma$). We have added the details to the paragraph.
>
> 7 In Line 290, if NSE is ablated, what’s the accuracy of AutoSC-GD on the full MNIST?
>
> $\textbf{Response}$:  If NSE is ablated, running the algorithm on the full MNIST is out-of-memory. We then use a computer with larger RAM to run the algorithm. The accuracy is 0.9882, close to the result on MNIST-30k reported in the paper.
>
> "The authors did not mention the limitation."
>
> $\textbf{Response}$: Thanks for pointing out.  We think one limitation is that we only studied automated spectral clustering while there are a few other clustering methods that are not based on similarity matrix. It can be a future work. We added the limitation to the conclusion section.

---

> > ### Comment · Reviewer_uu35 · 2022-08-08
> > **Feedback**
> >
> > Thanks for the authors' response. After reading the rebuttal, I have no further concerns and would like to keep my initial rating.

---

### Official Review · Reviewer_uYf6 · 2022-07-21

**Rating:** 5
**Confidence:** 4
**Soundness:** 2 fair
**Presentation:** 1 poor
**Contribution:** 3 good

**Summary:**

This paper presents a simple yet effective method for automated spectral clustering. In specific, the authors propose to find the most reliable affinity matrix via grid search or Bayesian optimization among a set of candidates given by different AMC methods with different hyper-parameters. Then, they propose a fast and accurate AMC method based on least squares representation and thresholding and prove its effectiveness theoretically. Finally, they extend the method to large-scale datasets.

**Questions:**

1. The second contribution is the proposal of Least Squares Representation (LSR) with Thresholding. However, LSR is commonly adopted in past decades and far from novel (Line 178 to 211).
2. The large-scale solution is fully shown in appendix with nothing in the manuscript. Please note that appendix is only a supplementary file to the manuscript. The readers have not responsibility to check out the appendix.
3. The manuscript format is out of order, such as Table 3.
4. From my opinion, the manuscript is not well organized. The authors want to emphasize Automated Spectral Clustering (AMC) which is imputed with a set of affinity matrices and outputs a selected one. So AMC is a process after the construction of affinity matrix (LSR in this paper). It means Section 4 is an application or showcase of the proposed method in section 3. But the authors pay a lot of attention on introducing LSR in section 4 and only leave a short sentence (line 195) to tell the connection between the two sections/points. This will easily confuse the reader.
5. In the abstract, the authors claim that the runtime is less than 90s. This is meaningless, since no one knows the runtime is recorded from any settings (which cpu, gpu, etc. ?). Please use computational complexity instead.
6. In table 3, why the runtimes of baselines, such as SSC and LRR,  are longer than the proposed AutoSC-GD and AutoSC-BO. In common sense, AutoSC-GD and AutoSC-BO requires extra time to search the optimal affinity matrix, leading to a longer runtime.
7. LSR, a very important baseline, is missing in experiment. Since the proposed method is developed upon LSR. So it is crucial to compare them to validate your add-on (AMC).
8. The limitations are not mentioned.

In conclusion, the proposed idea is interesting but the manuscript is far from prepared, including expression, organization and experiment validation.


**Strengths And Weaknesses:**

1. The idea is interesting to build the affinity matrix of spectral clustering by modeling its eigenvalue distribution.
2. The authors provide rich theories about the proposed criterion and method.
3. The large-scale problem is also considered.

---

> ### Author Response · Authors · 2022-07-29
> **We have re-organized the paper, improved the presentation, and added more numerical results.**
>
> We thank the reviewer for recognizing our contribution. Our response to your questions are as follows.
>
> 1 "...However, LSR is commonly adopted in past decades and far from novel (Line 178 to 211)."
>
> $\textbf{Response}$: Yes. LSR is not novel. Our contribution is to prove that LSR (as well as KLSR) with thresholding has the subspace detection property and empirically show (e.g., Figure 1, Figure 2, and Table 2) that they can do much better than many baselines such as SSC and BDR.
>
> 2 "The large-scale solution is fully shown in appendix with nothing in the manuscript..."
>
> $\textbf{Response}$:  Thanks for pointing it out. We now put the large-scale solution back to  the main paper (Section 3.5).
>
> 3 The manuscript format is out of order, such as Table 3.
>
> $\textbf{Response}$: Now we have fixed the problems.
>
> 4 "...But the authors pay a lot of attention on introducing LSR in section 4 and only leave a short sentence (line 195) to tell the connection between the two sections/points..."
>
> $\textbf{Response}$: Thanks for the comment. We then introduce LSR (the length has been reduced) in a new subsection of Section 3, i.e., Section 3.4, named as Discussion on AMC Methods and LSR with thresholding.  We put all numerical results in Section 4, the experiment section.
>
> We added the following content (last paragraph of Section 3.4) to show the connection between LSR and AutoSC:
>
> "Now we see that LSR and KLSR with thresholding can provide effective self-expressive affinity matrices for AutoSC without performing iterative optimization. On the other hand, the relative-eigen-gap is able to compare LSR with KLSR, compare different kernels, and evaluate $\lambda$, $\tau$, and kernel parameters. Note that if we use SSC and KSSC instead of LSR and KLSR, AutoSC will be very time consuming. If we use LSR and KLSR without threholding, AutoSC may not provide high clustering accuracy. We hope that AutoSC is not only automatic but also accurate and efficient."
>
> 5 "In the abstract...Please use computational complexity instead."
>
> $\textbf{Response}$:  Thank you for the suggestion. We have replaced it with the computational (time) complexity, which is linear with the number of data points. In Section 3.5, we added the details. The time complexity is $O(dmn+\tilde{d}s^2)$, where $d$ is the width of the neural network, $\tilde{d}$ depends on the specific affinity construction method, and $s\ll n$ denotes the number of landmarks.
>
> 6 "In table 3, why the runtimes of baselines, such as SSC and LRR, are longer than the proposed AutoSC-GD and AutoSC-BO..."
>
> $\textbf{Response}$:  In AutoSC-GD, we did not include the AMC methods requiring iterative optimization such as SSC and LRR because they are too slow (require hundreds of iterations and the per-iteration complexity is quadratic with $n$). Therefore, in AutoSC-GD, all AMC methods such as LSR have closed-form solutions. The time complexity of computing the relative eigen-gap is quadratic with $n$ and the number of trials of search are less than one hundred. As a result, the total time cost of AutoSC-GD is less than those of SSC and LRR.
>
> 7 "LSR, a very important baseline, is missing in experiment..."
>
> $\textbf{Response}$:  Actually, as mentioned in Footnote 4 of the previous version, the performance of LSR is not satisfactory. The reason is that LSR cannot produce disconnected graph without thresholding and it requires determining the parameter $\lambda$ carefully. The method EDSC presented in Table 4 (of the revised paper) is an improved version of LSR. It is outperformed by our method. Now we added the results of LSR. Here are some comparative results.
> \begin{equation*}
> \begin{matrix}\hline
> &\text{Yale B} & \text{ORL}& \text{AR} & \text{MNIST-1k}\\\\ \hline
> \text{LSR} &0.592 & 0.680&  0.665& 0.554\\\\
> \text{EDSC}& 0.806 & 0.712&0.673& 0.536\\\\
> \text{AutoSC-GD} & 0.897 & 0.795&0.786& 0.615\\\\ \hline
> \end{matrix}
> \end{equation*}
> These results of LSR have been added to Table 4 of the paper. The results of LSR on MNIST-10 k and Fashion-MNIST-10k have been added to Table 5 of the paper. LSR does not scale to the entire datasets of MNIST and Fashion-MNIST (out-of-memory). In all cases, our AutoSC outperformed LSR in terms of the clustering accuracy.
>
> 8 The limitations are not mentioned.
>
> $\textbf{Response}$: Thanks for reminding. One limitation is that we only studied automated spectral clustering while there are a few other clustering methods that are not based on similarity matrix. We added this limitation to the conclusion section.
>
> "In conclusion, the proposed idea is interesting but the manuscript is far from prepared, including expression, organization and experiment validation."
>
> $\textbf{Response}$: Thanks again for recognizing the novelty and contribution of our work. We have improved the organization and expression. We added more explanation (in the last paragraph of Section 3.4) about the connection between LSR and AutoSC. We also added the results of LSR to Table 4 and Table 5.

---

> > ### Comment · Reviewer_uYf6 · 2022-08-10
> > **Thanks for the rebuttal**
> >
> > The authors have addressed my concerns.

---

> > > ### Author Response · Authors · 2022-08-10
> > > **Thanks for recognizing our contribution**
> > >
> > > We sincerely appreciate your comments, suggestions, and increased score.

---

### Official Review · Reviewer_rahg · 2022-07-26

**Rating:** 6
**Confidence:** 3
**Soundness:** 3 good
**Presentation:** 3 good
**Contribution:** 3 good

**Summary:**

This paper proposes to find the most reliable affinity matrix for clustering using Bayesian optimization and grid search. The reliability is measured using the relative eigen-gap of the graph laplacian. Additionaly, a clustering method based on least squares is presented.

The main argument for proposing the new methodology is that a method for generating the affinity matrix for spectral clustering is still an open problem. Especially when one considers the hyperparameter tuning involved.

**Questions:**

Is there any connection between the "AMC using Least Squares Representation with Thresholding" and the relative eigen-gap method?

**Limitations:**

The paper has many different contributions and lacks real focus. The relative eigengap could be focused more on, and with removal of some of the other parts of the paper, the proofs could have been included to give a complete representation.

The connection to relatex work is quite limited as far as I can tell.

**Strengths And Weaknesses:**

Strengths
- A new method for comparing the eigenvalues of the Laplacian of an affinity matrix could influence the direction of spectral clustering research.
- The idea of direct comparison of different affinity matrix construction using grid search or bayesian optimization is interesting and could prove to be an efficient approach in real world clustering problems.
- The experiment section is extensive and shows that the method performs strongly.

Weaknesses:
- The papers is perhaps lacking in focus. Several different contributions are packed into the same paper, and it is not too obvious why they belong together. This leads to a confusing presentation which is sometimes quite difficult to read.
- The connection between the theoretical results and the contributions are not entirely clear in my opinion.
- The connection to current state of the art in clustering is somewhat lacking. It is fine to compare with similar methods of clustering (e.g. affinity based methods), but at least a mention of the best clustering method for the datasets used should be included.

---

> ### Author Response · Authors · 2022-07-29
> **The different contributions are closely related to each other and we have re-organized the paper and added more focus on AutoSC**
>
> We thank the reviewer for recognizing our contributions. We hope that the following responses can address your concerns and make you re-evaluate the quality of our work.
>
> $\textbf{On the weakness}$
>
> 1 "The papers is perhaps lacking in focus..."
>
> $\textbf{Response}$: As presented in page 2, our paper has the following four contributions:
>  $\textbf{(1) proposed the relative-eigen-gap based AutoSC}$; $\textbf{(2) applied Bayesian optimization to AutoSC}$; $\textbf{(3) proposed LSR and KLSR with thresholding for AutoSC}$; $\textbf{(4) proposed large-scale extension for AutoSC}$.
>
> These contributions are closely correlated to our topic AutoSC.  To ensure the clustering accuracy of AutoSC, we need a large search space that includes very effective methods of affinity matrix construction. We hope that AutoSC can select the best model from a wide rage of affinity matrix construction methods, rather than from a set of weak models such k-nn and Gaussian kernel. Although self-expressive methods such as SSC have shown significant improvement in constructing the affinity matrix for spectral clustering, they are often based on iterative optimization, which is very time consuming for AutoSC. Therefore, we proposed the LSR and KLSR with thresholding and proved their effectiveness theoretically.  As shown in Tables 2 and 4, the LSR and KLSR with thresholding are more effective than other methods.
>
> AutoSC has quadratic time and space complexity in evaluating every affinity matrix and hence is not applicable to big datasets. To solve the problem, we proposed the large-scale extension of AutoSC, which has linear time and space complexity and hence applicable to very large datasets. For example, shown in Tables 5 and 6, we can cluster the MNIST data in two minutes on an ordinary computer without using GPU.
>
> In sum, the LSR and KLSR with thresholding and the large-scale extension are important for AutoSC. They are resources that can be fully exploited by the relative-eigen-gap. We would like to make AutoSC not only automatic but also accurate and scalable.
>
> 2 "The connection between the theoretical results and the contributions are not entirely clear in my opinion."
>
> $\textbf{Response}$: We here further illustrate the connection between the theoretical results and the contributions:
>
> (1) $\textbf{Theorem 3.4}$ shows the effectiveness of the proposed relative-eigen-gap;
> (2) $\textbf{Theorem 3.7}$ verifies the effectiveness of the least squares representation with thresholding;
> (3) $\textbf{Proposition 3.8}$ shows that if the learned affinity matrix (e.g. given by the LSR with thresholding) has the subspace detection property and the relative-eigen-gap is nonzero, we can get a correct clustering result.
>
> 3 ``The connection to current state of the art ... at least a mention of the best clustering method for the datasets used should be included."
>
> $\textbf{Response}$: Thanks for pointing out this issue. On these datasets, deep learning methods have the SOTA performance. For example, on the Extended Yale B, the self-supervised learning based method proposed by [1] has an accuracy of 0.98. On ORL, the method proposed by [1] has an accuracy of 0.89. On Fashion-MNIST, the method proposed by [2] has an accuracy of 0.72.
> On MNIST, the method proposed by [3] has an accuracy of 0.99. On GTSRB, to the best of our knowledge, our method has SOTA performance. We added these explanation to the paper (lines 301-305). It is worth noting that our method outperformed many deep learning methods on MNIST and Fashion-MNIST.
>
> [1] Zhang et al. Self-Supervised Convolutional Subspace Clustering Network. CVPR 2019.
>
> [2] Zhang et al. Neural collaborative subspace clustering. ICML 2019.
>
> [3] Mahon and Lukasiewicz. Selective Pseudo-Label Clustering. arXiv 2021
>
>
>
> $\textbf{On the questions}$ "Is there any connection between the "AMC using Least Squares Representation with Thresholding" and the relative eigen-gap method?"
>
> $\textbf{Response}$: We provided Least Squares Representation with Thresholding as an efficient and accurate AMC method for AutoSC,  which can be evaluated by the relative eigen-gap.
>
> Because of the limit of characters, our responses to the "Limitations" are in the next section of the official comment.

---

> > ### Author Response · Authors · 2022-07-30
> > **Response to the "Limitations" and our detailed revision**
> >
> > $\textbf{On the limitation}$ "The paper has many different contributions and lacks real focus. The relative eigengap could be focused more on, and with removal of some of the other parts of the paper, the proofs could have been included to give a complete representation."
> > "The connection to relatex work is quite limited as far as I can tell."
> >
> > $\textbf{Response}$: Thank you for the suggestions, based on which we have made the following revisions.
> >
> > $\textbf{(1)}$ We reduced the content of the least square representation with thresholding and the large-scale extension of AutoSC in the main paper and put the discussion and some theoretical analysis into the supplementary material (e.g., Proposition A.1, Theorem C.3, and Proposition C.4).
> >
> > $\textbf{(2)}$ We have re-organized the paper thoroughly. Now Section 4 includes all experiments. Section 3 (AutoSC) has five subsections:
> > $\textbf{3.1 Preliminary Knowledge}$;
> > $\textbf{3.2 Relative Eigen-Gap Guided Search}$;
> > $\textbf{3.3 AutoSC via Bayesian Optimization}$;
> > $\textbf{3.4 Discussion on AMC Methods for AutoSC and LSR with Thresholding}$;
> > $\textbf{3.5 AutoSC+NSE for Large-Scale Data}$.
> >
> > Particularly, Section 3.4 discussed the limitations of existing affinity construction methods for AutoSC and introduced the LSR with thresholding and proved the effectiveness. LSR with thresholding can effectively improve the performance of AutoSC. Section 3.5 provided a large-scale extension for AutoSC. Then we can use AutoSC to cluster very large datasets efficiently.
> >
> > $\textbf{(3)}$ We added more focus on the relative eigen-gap in the main paper. For example, we added a new table, Table 3 in Section 4.1, to the main paper. The table compared our relative-eigen-gap reg$(\mathbf{L})$ with vanilla eigen-gap eg$(\mathbf{L})$ and the regularizer J$(\mathbf{L},\alpha)$ proposed by  [Meila and Shortreed,2006]. The results (also shown in the following table) showed that AutoSC with our reg$(\mathbf{L})$ outperformed AutoSC with eg$(\mathbf{L})$ and J$(\mathbf{L},\alpha)$. Note that J$(\mathbf{L},\alpha)$ introduced one more hyperparameter $\alpha$, which is difficult to determine.
> > \begin{equation}
> > \begin{matrix} \hline
> > & \text{YaleB} &\text{ORL}&\text{COIL20} &\text{AR} &\text{ MNIST}&\text{F-MNIST}\\\\ \hline
> > \text{eg}(\mathbf{L}) &{0.790}&{0.768} &{0.619}& 0.805&0.663&0.562\\\\
> > \text{J}(\mathbf{L},0) &{0.823}&\textbf{0.795}&0.750&0.804&0.726& 0.516\\\\
> > \text{J}(\mathbf{L},-0.1) &{0.818}&0.788&0.768&0.826&0.718&0.519\\\\
> > \text{J}(\mathbf{L},-1) &{0.812}&0.785 &0.769&0.817&0.722&0.523\\\\
> > \text{J}(\mathbf{L},-10) &{0.804}& 0.788&0.768& \textbf{0.832}&0.731&0.539\\\\
> > \text{J}(\mathbf{L},-100) &{0.788}& 0.765& 0.769 &0.817&0.735&0.545\\\\
> > \text{reg}(\mathbf{L}) &\textbf{0.897}&\textbf{0.795}&\textbf{0.782}&{0.786} &\textbf{0.755}&\textbf{0.595}\\\\ \hline
> > \end{matrix}
> > \end{equation}
> >
> > It is worth mentioning that in Section 4.1, Figure 1, Table 2, and Figure 2 are all about the relative-eigen-gap. For example, Table 2 verfified the effectiveness of the relative-eigen-gap in comparing different AMC methods including $\epsilon$-neighborhood, polynomial kernel, Gaussian kernel, KSSC (Gauss), KSSC (Poly), KLSR (Gauss), and KLSR (Poly). Figure 2 showed the effectiveness of the relative-eigen-gap in tuning the hyperparameters of SSC, KSSC, LSR, and KLSR. For instance, it is able to tune the kernel parameters such as the degree $q$ and bias $b$ of the polynomial kernel $k(\mathbf{x},\mathbf{y})=(\mathbf{x}^\top\mathbf{y}+b)^q$.
> >
> > $\textbf{(4)}$ In addition to the related work included in Section 2 (exploiting eigenvalue information for clustering and automated machine learning), we added more related work in Section 4, the experiment section. Specifically, we briefly introduced the paper with SOTA performances on the Extended Yale B, ORL, MNIST, Fashion-MNIST, and GTSRB in lines 301-305 (page 9). The added references are as follows.
> >
> > [1] Junjian Zhang, Chun-Guang Li, et al. Self-supervised convolutional subspace clustering network. In Proceedings of the IEEE/CVF conference on computer vision and pattern recognition, pages 5473–5482, 2019.
> >
> > [2] Tong Zhang, Pan Ji, Mehrtash Harandi, Wenbing Huang, and Hongdong Li. Neural collaborative subspace clustering. In International Conference on Machine Learning, pages 7384–7393. PMLR, 2019.
> >
> > [3] Louis Mahon and Thomas Lukasiewicz. Selective pseudo-label clustering. In German Conference on Artificial Intelligence (Kunstliche Intelligenz), pages 158–178. Springer, 2021.
> >
> > We are also glad to include more related work and discussion if the reviewer has any suggestion or recommendation.
> >
> > **Thank you again for your comments and evaluation.**

---

> > > ### Comment · Reviewer_rahg · 2022-08-10
> > > **Thank you for the thorough rebuttal**
> > >
> > > I thank the authors for a solid and thorough rebuttal. I believe that this has made the contributions much more clear and I am now convinced that this is a quite solid paper. I was initially a bit skeptic, but I see now that I was probably too strict. I will increase my score of the paper accordingly.

---

> > > > ### Author Response · Authors · 2022-08-10
> > > > **Thank you very much for recognizing our work.**
> > > >
> > > > The authors sincerely thank you for recognizing our work and increasing the score. You comments have made our paper stronger.

---

> ### Comment · Area_Chair_wVq4 · 2022-08-09
> **Author response phase closing today**
>
> The author-response phase closes today. Please acknowledge the author rebuttal and state if your position has changed. Thanks!

---

### Meta-Review · Area_Chair_wVq4 · 2022-08-29

**Recommendation:** Accept
**Confidence:** Less certain

**Metareview:**

The paper considers the problem of the choice of affinity matrix and associated hyperparameters for spectral clustering. The main contributions are a notion of relative eigen-gap to measure the quality of a clustering, the use of least squares representation with thresholding to construct the affinity matrix, and a novel method for large scale clustering. Extensive numerical results show the effectiveness of the proposed methods. In sum the paper provides an effective approach to automated hyperparameter selection for affinity-based spectral clustering.

**Award:**

No

---

### Decision · Program_Chairs · 2022-09-14

Accept